# Segment Anything in 3D with NeRFs

**Jiazhong Cen**[1][†]   **Zanwei Zhou**[1][†]   **Jiemin Fang**[2,3][‡]   **Chen Yang**[1],
**Wei Shen**[1][✉],   **Lingxi Xie**[2],   **Dongsheng Jiang**[2],   **Xiaopeng Zhang**[2],   **Qi Tian**[2]
[1] MoE Key Lab of Artificial Intelligence, AI Institute, Shanghai Jiao Tong University
[2] Huawei Inc. [3] School of EIC, Huazhong University of Science and Technology
`wei.shen@sjtu.edu.cn`

## Abstract

Recently, the Segment Anything Model (SAM) emerged as a powerful vision foundation model which is capable to segment anything in 2D images. This paper aims to generalize SAM to segment 3D objects. Rather than replicating the data acquisition and annotation procedure which is costly in 3D, we design an efficient solution, leveraging the Neural Radiance Field (NeRF) as a cheap and off-the-shelf prior that connects multi-view 2D images to the 3D space. We refer to the proposed solution as **SA3D**, for Segment Anything in 3D. It is only required to provide a manual segmentation prompt (*e.g.*, rough points) for the target object in a **single view**, which is used to generate its 2D mask in this view with SAM. Next, SA3D alternately performs **mask inverse rendering** and **cross-view self-prompting** across various views to iteratively complete the 3D mask of the target object constructed with voxel grids. The former projects the 2D mask obtained by SAM in the current view onto 3D mask with guidance of the density distribution learned by the NeRF; The latter extracts reliable prompts automatically as the input to SAM from the NeRF-rendered 2D mask in another view. We show in experiments that SA3D adapts to various scenes and achieves 3D segmentation within minutes. Our research reveals a potential methodology to lift the ability of a 2D vision foundation model to 3D, as long as the 2D model can steadily address promptable segmentation across multiple views. Our code is available at `https://github.com/Jumpat/SegmentAnythingin3D`.

## 1   Introduction

The computer vision community has been pursuing a vision foundation model that can perform basic tasks (*e.g.*, segmentation) in any scenario and for either 2D or 3D image data. Recently, the Segment Anything Model (SAM) [25] emerged and attracted a lot of attention, due to its ability to segment anything in 2D images, but generalizing the ability of SAM to 3D scenes remains mostly uncovered. One may choose to replicate the pipeline of SAM to collect and semi-automatically annotate a large set of 3D scenes, but the costly burden seems unaffordable for most research groups.

We realize that an alternative and efficient solution lies in equipping the 2D foundation model (*i.e.*, SAM) with 3D perception via a 3D representation model. In other words, there is no need to establish a 3D foundation model from scratch. However, there is a prerequisite: the 3D representation model shall be capable to render 2D views and register 2D segmentation results to the 3D scene. Thus, we use the Neural Radiance Fields (NeRF) [38, 53, 3] as an off-the-shelf solution. NeRF is a family of algorithms that formulates each 3D scene into a deep neural network that serves as a 3D prior connecting multiple 2D views.

---

[✉]Corresponding author.
[†]Equal contribution.
[‡]Project lead.

37th Conference on Neural Information Processing Systems (NeurIPS 2023).

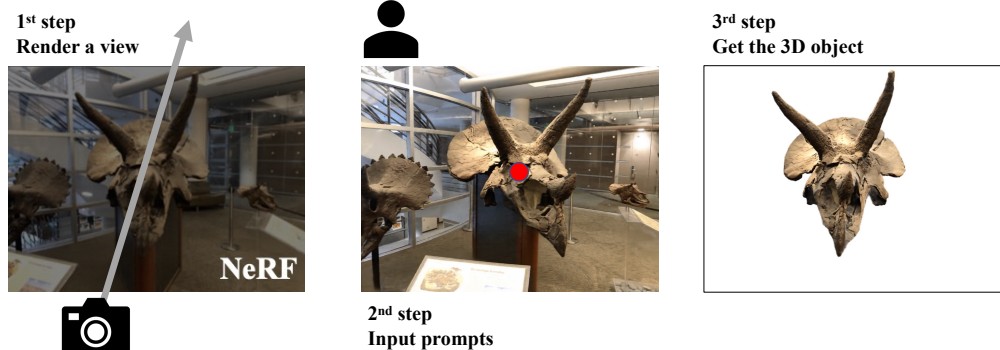

**1st step**
Render a view

NeRF

**2nd step**
Input prompts

**3rd step**
Get the 3D object

Figure 1: Given any pre-trained NeRF, SA3D takes prompts from one single rendered view as input and outputs the 3D segmentation result for the specific target.

As shown in Figure 1, our solution is named Segment Anything in 3D (**SA3D**). Given a NeRF trained on a set of 2D images, SA3D takes prompts (*e.g.*, click points on the object) in a single rendered view as input, which is used to generate a 2D mask in this view with SAM. Next, SA3D alternately performs two steps across various views to iteratively complete the 3D mask of the object constructed with voxel grids. In each round, the first step is **mask inverse rendering**, in which the previous 2D segmentation mask obtained by SAM is projected onto the 3D mask via density-guided inverse rendering offered by the NeRF. The second step is **cross-view self-prompting**, in which NeRF is used to render the 2D segmentation mask (which may be inaccurate) based on the 3D mask and the image from another view, then a few point prompts are automatically generated from the rendered mask and fed into SAM to produce a more complete and accurate 2D mask. The above procedure is executed iteratively until all necessary views have been sampled.

We conduct various (*e.g.*, object, part-level) segmentation tasks on the Replica [51] and NVOS [47] datasets. Without re-training/re-designing SAM or NeRF, SA3D easily and efficiently adapts to different scenarios. Compared to existing approaches, SA3D enjoys a simplified pipeline that typically completes 3D segmentation within minutes. SA3D not only offers an efficient tool for segmenting anything in 3D, but also reveals a generic methodology to lift 2D foundation models to the 3D space. The only prerequisite lies in the ability to steadily address promptable segmentation across multiple views, and we hope it becomes a general property of 2D foundation models in the future.

## 2  Related Work

**2D Segmentation**    Since FCN [36] has been proposed, research on 2D image segmentation has experienced a rapid growth. Various sub-fields of segmentation have been explored deeply by numerous studies [18, 24, 4, 71]. With transformers [58, 10] entering the field of segmentation, many new segmentation architectures [72, 7, 6, 52, 63] have been proposed and the whole field of segmentation has been further developed. A recent significant breakthrough in this field is the Segment Anything Model (SAM) [25]. As an emerging vision foundation model, SAM is recognized as a potential game-changer, which aims to unify the 2D segmentation task by introducing a prompt-based segmentation paradigm. An analogous model to SAM is SEEM [75], which also exhibits impressive open-vocabulary segmentation capabilities.

**3D Segmentation**    Numerous methods have explored various types of 3D representations to perform 3D segmentation. These scene representations include RGB-D images [60, 62, 64, 8], point clouds [44, 45, 70] and grid space such as voxels [21, 55, 35], cylinders [74] and bird's eye view space [67, 16]. Although 3D segmentation has been developed for a period of time, compared with 2D segmentation, the scarcity of labeled data and high computational complexity make it difficult to design a unified framework similar to SAM.

**Lifting 2D Vision Foundation Models to 3D**    To tackle the limitation of data scarcity, many previous studies [23, 43, 9, 17, 69, 31, 65, 22] explored lifting 2D foundation models to 3D. In these studies, the most relevant work to SA3D is LERF [23], which trains a feature field of the Vision-Language Model (*i.e.*, CLIP [46]) together with the radiance field. Compared with SA3D,

LERF focuses on coarsely localizing the specific objects with text prompts but not fine-grained 3D segmentation. The reliance on CLIP features makes it insensitive to the specific location information of the target object. When there are multiple objects with similar semantics in the scene, LERF cannot perform effective 3D segmentation. The remaining methods mainly focus on point clouds. By connecting the 3D point cloud with specific camera poses with 2D multi-view images, the extracted features by 2D foundation models can be projected to the 3D point cloud. The data acquisition of these methods is more expensive than ours, *i.e.*, acquiring multi-view images for NeRFs.

**Segmentation in NeRFs**   Neural Radiance Fields (NeRFs) [38, 53, 3, 1, 40, 19, 13, 61, 30, 12] are a series of 3D implicit representation. Inspired by their success in 3D consistent novel view synthesis, numerous studies have delved into the realm of 3D segmentation within NeRFs. Zhi *et al.* [73] proposes Semantic-NeRF, a method that incorporates semantics into appearance and geometry. They showcase the potential of NeRFs in label propagation and refinement. NVOS [47] introduces an interactive approach to select 3D objects from NeRFs by training a lightweight multi-layer perception (MLP) using custom-designed 3D features. Other approaches, *e.g.* N3F [57], DFF [27], LERF [23] and ISRF [15], aim to lift 2D visual features to 3D through training additional feature fields. These methods are required to re-design/-train NeRF models and usually involve additional feature-matching processes. There are also some other instance segmentation and semantic segmentation approaches [50, 41, 11, 68, 34, 20, 2, 14, 59, 28] combined with NeRFs.

The most closely related approach to our SA3D is MVSeg [39], a component of SPIn-NeRF [39], which focuses on NeRF inpainting. MVSeg adopts video segmentation techniques to propagate a 2D mask across different views and employs these masks as labels for training a Semantic-NeRF model. However, video segmentation models lack explicit 3D structure information, which are hard to handle significant occlusions in complex scenes. Our method aims at building NeRF-driven consistency across views based on self-prompting and lifting 2D masks to robust 3D masks.

## 3   Method

In this section, we first give a brief review of Neural Radiance Fields (NeRFs) and the Segment Anything Model (SAM). Then we introduce the overall pipeline of SA3D. Finally, we demonstrate the design of each component in SA3D in detail.

### 3.1   Preliminaries

**Neural Radiance Fields (NeRFs)**   Given a training dataset $\mathcal{I}$ of multi-view 2D images, NeRFs [38] learn a function $f_{\boldsymbol{\theta}} : (\mathbf{x}, \mathbf{d}) \rightarrow (\mathbf{c}, \sigma)$, which maps the spatial coordinates $\mathbf{x} \in \mathbb{R}^3$ and the view direction $\mathbf{d} \in \mathbb{S}^2$ of a point into the corresponding color $\mathbf{c} \in \mathbb{R}^3$ and volume density $\sigma \in \mathbb{R}$. $\boldsymbol{\theta}$ denotes the learnable parameters of the function $f$ which is usually represented by a multi-layer perceptron (MLP). To render an image $\mathbf{I}_{\boldsymbol{\theta}}$, each pixel undergoes a ray casting process where a ray $\mathbf{r}(t) = \mathbf{x}_o + t\mathbf{d}$ is projected through the camera pose. Here, $\mathbf{x}_o$ is the camera origin, $\mathbf{d}$ is the ray direction, and $t$ denotes the distance of a point along the ray from the origin. The RGB color $\mathbf{I}_{\boldsymbol{\theta}}(\mathbf{r})$ at the location determined by ray $\mathbf{r}$ is obtained via a differentiable volume rendering algorithm:

$$\mathbf{I}_{\boldsymbol{\theta}}(\mathbf{r}) = \int_{t_n}^{t_f} \omega(\mathbf{r}(t))\mathbf{c}(\mathbf{r}(t), \mathbf{d})\mathrm{d}t, \tag{1}$$

where $\omega(\mathbf{r}(t)) = \exp(- \int_{t_n}^{t} \sigma(\mathbf{r}(s))\mathrm{d}s) \cdot \sigma(\mathbf{r}(t))$, and $t_n$ and $t_f$ denote the near and far bounds of the ray, respectively.

**Segment Anything Model (SAM)**   SAM [25] takes an image $\mathbf{I}$ and a set of prompts $\mathcal{P}$ as input, and outputs the corresponding 2D segmentation mask $\mathbf{M}_{\texttt{SAM}}$ in the form of a bitmap, *i.e.,*

$$\mathbf{M}_{\texttt{SAM}} = s(\mathbf{I}, \mathcal{P}). \tag{2}$$

The prompts $\mathbf{p} \in \mathcal{P}$ can be points, boxes, texts, and masks.

### 3.2   Overall Pipeline

We assume that we already have a NeRF model trained on the dataset $\mathcal{I}$. Throughout this paper, unless otherwise specified, we opt to employ the TensoRF [3] as the NeRF model, considering its

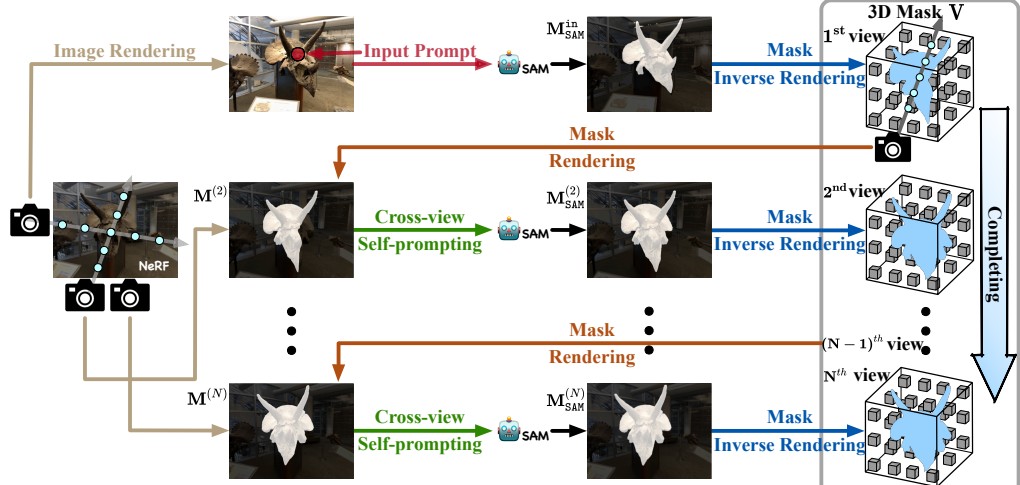

Figure 2: The overall pipeline of SA3D. Given a NeRF trained on a set of multi-view 2D images, SA3D first takes prompts in a single view for the target object as input and uses SAM to produce a 2D mask in this view with these prompts. Then, SA3D performs an alternated process of **mask inverse rendering** and **cross-view self-prompting** to complete the 3D mask of the target object constructed with voxel grids. Mask inverse rendering is performed to project the 2D mask obtained by SAM onto the 3D mask according to the learned density distribution embedded in the NeRF. Cross-view self-prompting is conducted to extract reliable prompts automatically as the input to SAM from the NeRF-rendered 2D mask given a novel view. This alternated process is executed iteratively until we get the complete 3D mask.

superior efficiency of training and rendering. As shown in Figure 2, an image $\mathbf{I}^{\text{in}}$ from a specific view is first rendered with the pre-trained NeRF model. A set of prompts (*e.g.*, in this paper, we often use a set of points), $\mathcal{P}^{\text{in}}$, is introduced and fed into SAM along with the rendered image. The 2D segmentation mask $\mathbf{M}^{\text{in}}_{\text{SAM}}$ of the according view is obtained, which is then projected onto the 3D mask $\mathbf{V} \in \mathbb{R}^3$ constructed voxel grids with the proposed **mask inverse rendering** technique (Section 3.3). Then a 2D segmentation mask $\mathbf{M}^{(n)}$ from a novel view is rendered from the 3D mask. The rendered mask is usually inaccurate. We propose a **cross-view self-prompting** method (Section 3.4) to extract point prompts $\mathcal{P}^{(n)}$ from the rendered mask and further feed them into SAM. Thus a more accurate 2D mask $\mathbf{M}^{(n)}_{\text{SAM}}$ in this novel view is produced and also projected onto the voxel grids to complete the 3D mask. The above procedure is executed iteratively with more views traversed. Meanwhile, the 3D mask become more and more complete. The whole process bridges 2D segmentation results with 3D ones efficiently. Noting that no neural network needs to be optimized except the 3D mask.

### 3.3 Mask Inverse Rendering

As shown in Equation (1), the color of each pixel in a rendered image is determined by a sum of weighted colors along the corresponding ray. The weight $\omega(\mathbf{r}(t))$ reveals the object structure within the 3D space, where a high weight indicates the corresponding point close to the object's surface. Mask inverse rendering aims to project the 2D mask to the 3D space to form the 3D mask based on these weights.

Formally, the 3D mask is represented as voxel grids $\mathbf{V} \in \mathbb{R}^{L \times W \times H}$, where each grid vertex stores a zero-initialized soft mask confidence score. Based on these voxels grids, each pixel of the 2D mask from one view is rendered as

$$\mathbf{M}(\mathbf{r}) = \int_{t_n}^{t_f} \omega(\mathbf{r}(t))\mathbf{V}(\mathbf{r}(t))\mathrm{d}t, \tag{3}$$

where $\mathbf{r}(t)$ is the ray casting through the mask pixel, $\omega(\mathbf{r}(t))$ is inherited from density values of the pre-trained NeRF, and $\mathbf{V}(\mathbf{r}(t))$ denotes the mask confidence score at the location $\mathbf{r}(t)$ obtained from voxel grids $\mathbf{V}$[1]. Denote $\mathbf{M}_{\text{SAM}}(\mathbf{r})$ as the corresponding mask generated by SAM. When $\mathbf{M}_{\text{SAM}}(\mathbf{r}) = 1$,

---

[1]$\mathbf{V}(\mathbf{r}(t))$ is computed by trilinearly interpolating vertex values of the 3D mask grids.

the goal of mask inverse rendering is to increase $\mathbf{V}(\mathbf{r}(t))$ with respect to $\omega(\mathbf{r}(t))$. In practice, this can be optimized using the gradient descent algorithm. For this purpose, we define the mask projection loss as the negative product between $\mathbf{M}_{\text{SAM}}(\mathbf{r})$ and $\mathbf{M}(\mathbf{r})$:

$$\mathcal{L}_{\text{proj}} = - \sum_{\mathbf{r} \in \mathcal{R}(\mathbf{I})} \mathbf{M}_{\text{SAM}}(\mathbf{r}) \cdot \mathbf{M}(\mathbf{r}), \tag{4}$$

where $\mathcal{R}(\mathbf{I})$ denotes the ray set of the image $\mathbf{I}$.

The mask projection loss is constructed based on the assumption that both the geometry from the NeRF and the segmentation results of SAM are accurate. However, in practice, this is not always the case. We append a negative refinement term to the loss to optimize the 3D mask grids according to multi-view mask consistency:

$$\mathcal{L}_{\text{proj}} = - \sum_{\mathbf{r} \in \mathcal{R}(\mathbf{I})} \mathbf{M}_{\text{SAM}}(\mathbf{r}) \cdot \mathbf{M}(\mathbf{r}) + \lambda \sum_{\mathbf{r} \in \mathcal{R}(\mathbf{I})} (1 - \mathbf{M}_{\text{SAM}}(\mathbf{r})) \cdot \mathbf{M}(\mathbf{r}), \tag{5}$$

where $\lambda$ is a hyper-parameter to determine the magnitude of the negative term. With this negative refinement term, only if SAM consistently predicts a region as foreground from different views, SA3D marks its corresponding 3D region as foreground. In each iteration, the 3D mask $\mathbf{V}$ is updated via $\mathbf{V} \leftarrow \mathbf{V} - \eta \frac{\partial \mathcal{L}_{\text{proj}}}{\partial \mathbf{V}}$ with gradient descent, where $\eta$ denotes the learning rate.

### 3.4 Cross-view Self-prompting

Mask inverse rendering enables projecting 2D masks into the 3D space to form the 3D mask of a target object. To construct accurate 3D mask, substantial 2D masks from various views need to be projected. SAM can provide high-quality segmentation results given proper prompts. However, manually selecting prompts from every view is time-consuming and impractical. We propose a cross-view self-prompting mechanism to produce prompts for different novel views automatically.

Specifically, we first render a novel-view 2D segmentation mask $\mathbf{M}^{(n)}$ from the 3D mask grids $\mathbf{V}$ according to Equation (3). This mask is usually inaccurate, especially at the preliminary iteration of SA3D. Then we obtain some point prompts from the rendered mask with a specific strategy. The above process is named cross-view self-prompting. While there are multiple possible solutions for this strategy, we present a feasible one that has been demonstrated to be effective.

**Self-prompting Strategy** Given an inaccurate 2D rendered mask $\mathbf{M}^{(n)}$, the self-prompting strategy aims to extract a set of prompt points $\mathcal{P}_s$ from it, which can help SAM to generate 2D segmentation result as accurate as possible. It is important to note that $\mathbf{M}^{(n)}$ is not a typical 2D bitmap, but rather a confidence score map computed using Equation (3). Since each image pixel $\mathbf{p}$ corresponds to a ray $\mathbf{r}$ in a rendered view, we use $\mathbf{p}$ for an easier demonstration of the prompt selection strategy on images.

As $\mathcal{P}_s$ is initialized to an empty set, the first prompt point $\mathbf{p}_0$ is selected as the point with the highest mask confidence score: $\mathbf{p}_0 = \arg \max_{\mathbf{p}} \mathbf{M}^{(n)}(\mathbf{p})$. To select new prompt points, we first mask out square shaped regions[2] on $\mathbf{M}^{(n)}$ centered with each existing point prompt $\hat{\mathbf{p}} \in \mathcal{P}_s$. Considering the depth $z(\mathbf{p})$ can be estimated by the pre-trained NeRF, we transform the 2D pixel $\mathbf{p}$ to the 3D point $\mathcal{G}(\mathbf{p}) = (x(\mathcal{G}(\mathbf{p})), y(\mathcal{G}(\mathbf{p})), z(\mathcal{G}(\mathbf{p})))$:

$$\begin{pmatrix} x(\mathcal{G}(\mathbf{p})) \\ y(\mathcal{G}(\mathbf{p})) \\ z(\mathcal{G}(\mathbf{p})) \end{pmatrix} = z(\mathbf{p}) \mathbf{K}^{-1} \begin{pmatrix} x(\mathbf{p}) \\ y(\mathbf{p}) \\ 1 \end{pmatrix} \tag{6}$$

where $x(\mathbf{p}), y(\mathbf{p})$ denote the 2D coordinates of $\mathbf{p}$, and $\mathbf{K}$ denotes the camera intrinsics. The new prompt point is expected to have a high confidence score while being close to existing prompt points. Considering the two factors, we introduce a decay term to the confidence score. Let $d(\cdot, \cdot)$ denote the min-max normalized Euclidean distance. For each remaining point $\mathbf{p}$ in $\mathbf{M}^{(n)}$, the decay term is

$$\Delta \mathbf{M}^{(n)}(\mathbf{p}) = \min\{\mathbf{M}^{(n)}(\hat{\mathbf{p}}) \cdot d(\mathcal{G}(\mathbf{p}), \mathcal{G}(\hat{\mathbf{p}})) \mid \hat{\mathbf{p}} \in \mathcal{P}_s\}. \tag{7}$$

Then a decayed mask confidence score $\tilde{\mathbf{M}}^{(n)}(\mathbf{p})$ is computed as

$$\tilde{\mathbf{M}}^{(n)}(\mathbf{p}) = \mathbf{M}^{(n)}(\mathbf{p}) - \Delta \mathbf{M}^{(n)}(\mathbf{p}). \tag{8}$$

---

[2] The side length of the region is set as the radius of a circle whose area is equal to the rendered mask $\mathbf{M}^{(n)}$.

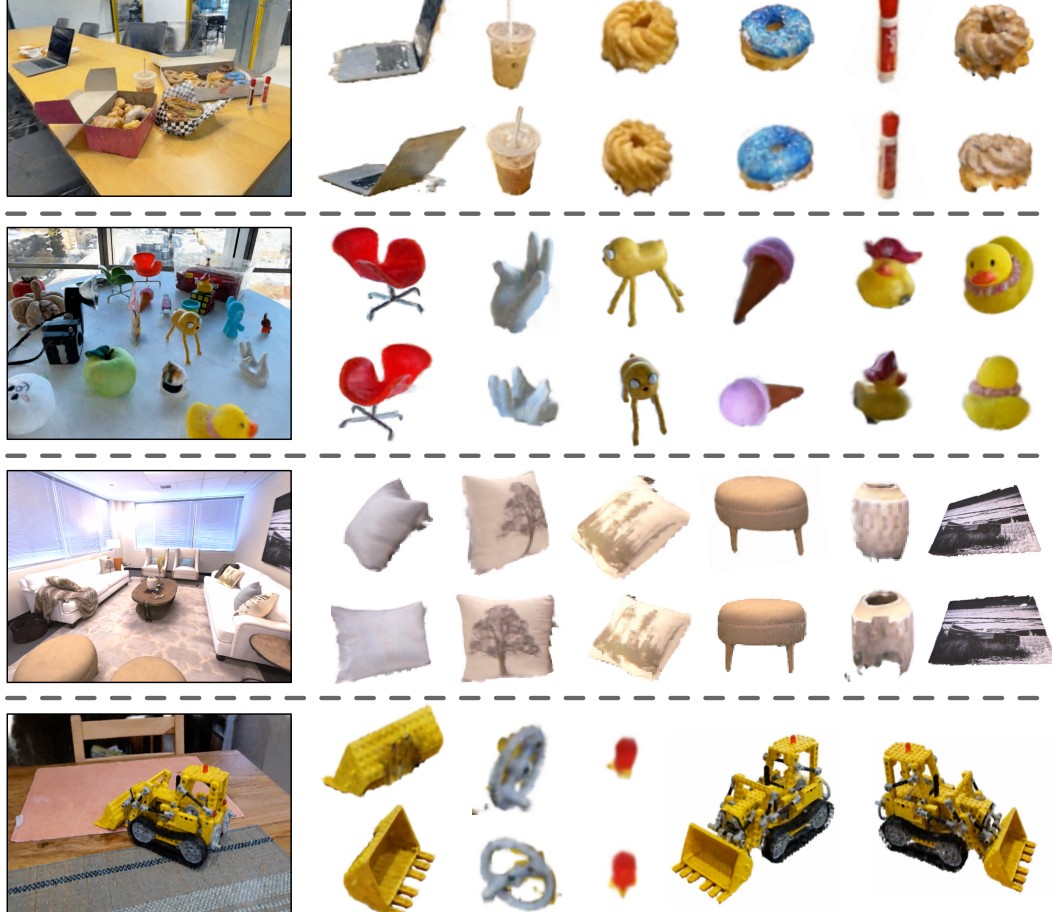

Figure 3: Some visualization results in different scenes (LERF-donuts [23], LERF-figurines, Replica-room0 [51] and 360-kitchen [1]).

The remaining point with the highest decayed score, *i.e.*, $\mathbf{p}^* = \arg\max_{\mathbf{p} \notin \mathcal{P}_s} \tilde{\mathbf{M}}^{(n)}(\mathbf{p})$, is added to the prompt set: $\mathcal{P}_s = \mathcal{P}_s \cup \{\mathbf{p}^*\}$. The above selection process is repeated until either the number of prompts $|\mathcal{P}_s|$ reaches a predefined threshold $n_p$ or the maximum value of $\tilde{\mathbf{M}}^{(n)}(\mathbf{p})$ is smaller than 0.

**IoU-aware View Rejection** When the target object is rendered in heavily occluded views, SAM may produce incorrect segmentation results and degrades the quality of the 3D mask. To avoid such situations, we introduce an additional view rejection mechanism based on the intersection-over-union (IoU) between the rendered mask $\mathbf{M}^{(n)}$ and the SAM prediction $\mathbf{M}_{\text{SAM}}^{(n)}$. If the IoU falls below a predefined threshold $\tau$, it indicates a poor overlap between the two masks. The prediction from SAM is rejected, and the mask inverse rendering step is skipped in this iteration.

## 4 Experiments

In this section, we quantitatively evaluate the segmentation ability of SA3D on various datasets. Then, we qualitatively demonstrate the versatility of SA3D, which can conduct instance segmentation, part segmentation, and text-prompted segmentation *etc*.

### 4.1 Datasets

For quantitative experiments, we use the Neural Volumetric Object Selection (NVOS) [47], SPIn-NeRF [39], and Replica [51] datasets. The NVOS [47] dataset is based on the LLFF dataset [37], which includes several forward-facing scenes. For each scene, NVOS provides a reference view

with scribbles and a target view with 2D segmentation masks annotated. Similar to NVOS, SPIn-NeRF [39] annotates some data manually to evaluate interactive 3D segmentation performance. These annotations are based on some widely-used NeRF datasets [37, 38, 29, 26, 13]. The Replica [51] dataset provides high-quality reconstruction ground truths of various indoor scenes, including clean dense geometry, high-resolution and high-dynamic-range textures, glass and mirror surface information, semantic classes, planar segmentation, and instance segmentation masks. For qualitative analysis, we use the LLFF [37] dataset and the 360° dataset [1]. SA3D is further applied to the LERF [23] dataset, which contains more realistic and challenging scenes.

## 4.2  Quantitative Results

**NVOS Dataset**  For fair comparisons, we follow the experimental setting of the original NVOS [47]. We first scribble on the reference view (provided by the NVOS dataset) to conduct 3D segmentation, and then render the 3D segmentation result on the target view and evaluate the IoU and pixel-wise accuracy with the provided ground truth. Note that the scribbles are preprocessed to meet the requirements of SAM. More details can be found in the supplement.

Table 1: Quantitative results on NVOS.

| Method | mIoU (%) | mAcc (%) |
|---|---|---|
| Graph-cut (3D) [48, 47] | 39.4 | 73.6 |
| NVOS [47] | 70.1 | 92.0 |
| ISRF [15] | 83.8 | 96.4 |
| SA3D (ours) | **90.3** | **98.2** |

As shown in Table 1, SA3D outperforms previous approaches by large margins, *i.e.*, +6.5 mIoU over the previous SOTA ISRF and +20.2 mIoU over the NVOS.

Table 2: Quantitative results on the SPIn-NeRF dataset.

| Scenes | Single view | | MVSeg [39] | | SA3D (ours) | |
|---|---|---|---|---|---|---|
| | IoU (%) | Acc (%) | IoU (%) | Acc (%) | IoU (%) | Acc (%) |
| Orchids | 79.4 | 96.0 | **92.7** | 98.8 | 83.6 | 96.9 |
| Leaves | 78.7 | 98.6 | 94.9 | 99.7 | **97.2** | 99.9 |
| Fern | 95.2 | 99.3 | 94.3 | 99.2 | **97.1** | 99.6 |
| Room | 73.4 | 96.5 | **95.6** | 99.4 | 88.2 | 98.3 |
| Horns | 85.3 | 97.1 | 92.8 | 98.7 | **94.5** | 99.0 |
| Fortress | 94.1 | 99.1 | 97.7 | 99.7 | **98.3** | 99.8 |
| Fork | 69.4 | 98.5 | 87.9 | 99.5 | **89.4** | 99.6 |
| Pinecone | 57.0 | 92.5 | **93.4** | 99.2 | 92.9 | 99.1 |
| Truck | 37.9 | 77.9 | 85.2 | 95.1 | **90.8** | 96.7 |
| Lego | 76.0 | 99.1 | 74.9 | 99.2 | **92.2** | 99.8 |
| mean | 74.6 | 95.5 | 90.9 | 98.9 | **92.4** | 98.9 |

**SPIn-NeRF Dataset**  We follow SPIn-NeRF [39] to conduct label propagation for evaluation. Given a specific reference view of a target object, the 2D ground-truth mask of this view is available. The prompt input operation is omitted, while the 2D ground-truth mask of the target object from the reference view is directly used for the initialization of the 3D mask grids. This is reasonable since in most situations users can refine their input prompts to help SAM generate a 2D mask as accurately as possible from the reference view. Once the 3D mask grids are initialized, the subsequent steps are exactly the same as described in Section 3. With the 3D mask grids finalized, the 2D masks in other views are rendered to calculate the IoU with the 2D ground-truth masks. Results can be found in Table 2. SA3D is demonstrated to be superior in both forward-facing and 360° scenes.

In Tables 2 and 3, "Single view" refers to conducting mask inverse rendering exclusively for the 2D ground-truth mask of the reference view. This process is equivalent to mapping the 2D mask to the 3D space based on the corresponding depth information, without any subsequent learnable/updating step. We present these results to demonstrate the gain of the alternated process in our framework. As shown in Table 2, SA3D outperforms MVSeg [39] in most scenes, especially +5.6 mIoU on Truck and +17.3 mIoU on Lego. Besides, compared with the "Single view" model, a significant promotion is achieved, *i.e.* +17.8 mIoU, which further proves the effectiveness of our method.

Table 3: Quantitative results on Replica (mIoU).

| Scenes | office0 | office1 | office2 | office3 | office4 | room0 | room1 | room2 | mean |
|---|---|---|---|---|---|---|---|---|---|
| Single view | 68.7 | 56.5 | 68.4 | 62.2 | 57.0 | 55.4 | 53.8 | 56.7 | 59.8 |
| MVSeg [39] | 31.4 | 40.4 | 30.4 | 30.5 | 25.4 | 31.1 | 40.7 | 29.2 | 32.4 |
| SA3D (ours) | **84.4** | **77.0** | **88.9** | **84.4** | **82.6** | **77.6** | **79.8** | **89.2** | **83.0** |

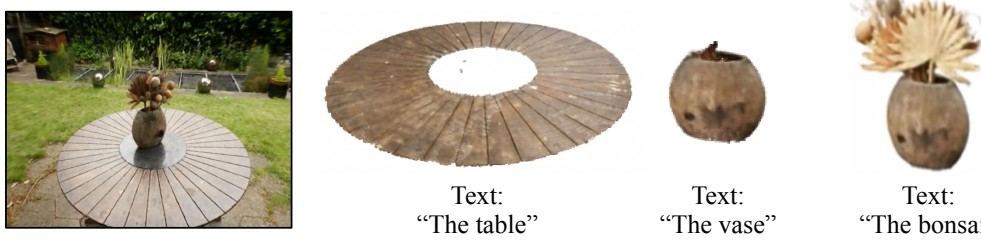

Text:          Text:          Text:
"The table"    "The vase"    "The bonsai"

Figure 4: 3D segmentation results of SA3D with the text prompts in 360-garden [1].

**Replica Dataset**  We use the processed Replica data with 2D instance labels provided by Zhi *et al.* [73] to evaluate the segmentation performance of SA3D. We retrieve all views containing each object and specify one reference view. With a similar setting of experiments on the SPIn-NeRF dataset, we use the ground-truth mask of the reference view and perform SA3D to conduct label propagation for evaluation. For each scene in Replica, around 20 objects are chosen for evaluation. Refer to the supplement for more details. As shown in Table 3, the mean IoU (mIoU) is reported for all available objects in different scenes. We exclude the pixel-wise accuracy metric since an object only appears in a few views in the indoor scenes of Replica, where the pixel-wise accuracy is too high to serve as a reliable metric.

In complex indoor scenes of Replica, MVSeg's strategy based on video segmentation proves to be ineffective, which generates numerous inaccurate 2D pseudo-labels, even using the Semantic-NeRF [73] for refinement. Consequently, the final 3D segmentation results of MVSeg even underperform those achieved by the "Single view" method. In contrast, SA3D accurately captures segmented objects in complex 3D scenes. Visualization results are shown in Figure 3.

### 4.3 Qualitative Results

We conduct three kinds of segmentation tasks: object segmentation, part segmentation and text-prompting segmentation. The first two are the core functions of SA3D. As shown in Figure 3, SA3D demonstrates its capability to segment diverse 3D objects across different scenes, even when the objects are of small scales. Besides, SA3D can also handle challenging part segmentation. The last row of the figure showcases SA3D's precise segmentation of the bucket, small wheel, and dome light of the Lego bulldozer. Figure 4 demonstrates the potential of SA3D in combining with language models. Given a text phrase, the corresponding object can be accurately cut out. The text-prompting segmentation is built upon Grounding-DINO [33], a model capable of generating bounding boxes for objects based on text prompts. These bounding boxes serve as input prompts for SA3D in the segmentation process.

Table 4: Ablation on different numbers of views for 3D mask generation. Numbers in parentheses represent the view percentage of total training views.

| Number of Views | 5 (10%) | 9 (20%) | 21 (50%) | 43 (100%) |
|---|---|---|---|---|
| IoU on Fortress (forward facing) | 97.8 | 98.3 | 98.3 | 98.3 |
| Time Cost (s) | 7.6 | 12.8 | 29.0 | 59.0 |

| Number of Views | 11 (10%) | 21 (20%) | 51 (50%) | 103 (100%) |
|---|---|---|---|---|
| IoU on Lego (360°) | 84.5 | 84.8 | 91.5 | 92.2 |
| Time Cost (s) | 23.5 | 43.5 | 103.8 | 204.9 |

### 4.4 Ablation Study

**Number of Views** The process of mask inverse rendering and cross-view self-prompting is alternated across different views. By default, we utilize all available views in the training set $\mathcal{I}$. However, to expedite the 3D segmentation procedure, the number of views can be reduced. As shown in Table 4, We perform experiments on two representative scenes from the SPIn-NeRF [39] dataset to demonstrate this characteristic. The views are uniformly sampled from the sorted training set. In forward facing scenes where the range of the camera poses is limited, satisfactory results can be achieved by selecting only a few views. On an Nvidia RTX 3090 GPU, the 3D segmentation process with 5 views can be completed within 10 seconds. On the contrary, in scenes where the range of camera poses is wider, a larger number of views are required to yield greater improvements. Note that even with 50 views, the segmentation task can still be completed in less than 2 minutes.

**Hyper-parameters** SA3D involves three hyper-parameters: the IoU rejection threshold $\tau$, the loss balance coefficient $\lambda$ in Equation (5), and the number of self-prompting points $n_p$. As shown in Table 6, too small $\tau$ values lead to unstable SAM predictions, introducing noises to the 3D mask; too large $\tau$ values impede the 3D mask from getting substantial information. Table 7 indicates slightly introducing a negative term with the $\lambda$ factor can reduce noise for mask projection. However, a too-large negative term may make the mask completion process unstable and causes degraded performance. The selection of $n_p$ depends on the specific segmentation target, as SAM tends to produce over-segmented results that capture finer details of objects. As shown in Figure 5, for objects with a relatively large scale and complex structures, a bigger $n_p$ produces better results. Empirically. setting $n_p$ to 3 can meet the requirements of most situations.

**Self-prompting Strategy** Without the 3D distance based confidence decay (Equation (7)), our self-prompting strategy degrades to a simple 2D NMS (Non-Maximum Suppression), which selects a prompt point with the highest confidence score and then masks out a region around it. To showcase the efficacy of our design, we conduct experiments using the NVOS benchmark and presented per-scene results for in-depth analysis.

Table 5: Ablation on the confidence decay term of the self-prompting strategy.

| Scenes | w/ Confidence Decay Term | | w/o Confidence Decay Term | |
| --- | --- | --- | --- | --- |
| | IoU (%) | Acc (%) | IoU (%) | Acc (%) |
| Fern | 82.9 | 94.4 | 82.9 | 94.4 |
| Flower | 94.6 | 98.7 | 94.6 | 99.7 |
| Fortress | 98.3 | 99.7 | 98.4 | 99.7 |
| Horns (center) | 96.2 | 99.3 | 96.2 | 99.3 |
| Horns (Left) | 90.2 | 99.4 | 88.8 | 99.3 |
| Leaves | 93.2 | 99.6 | 93.2 | 99.6 |
| Orchids | 85.5 | 97.3 | 85.4 | 97.3 |
| Trex | 82.0 | 97.4 | 64.0 | 93.3 |
| mean | 90.3 | 98.2 | 87.9 | 97.7 |

Table 5 shows that a simple NMS self-prompting is enough for most cases. But for hard cases like 'LLFF-trex' (a trex skeleton, as shown in Figure 5), where a large number of depth jumps, the confidence decay term contributes a lot. In such a situation, inaccurate masks bleed through gaps in the foreground onto the background. If the self-prompting mechanism generates prompts on these inaccurate regions, SAM may produce plausible segmentation results that can cheat the IoU-rejection mechanism and finally the segmentation results will involve unwanted background regions.

**2D Segmentation Models** In addition to SAM, we also incorporate four other prompt-based 2D segmentation models [75, 49, 32, 5] into our framework to demonstrate the generalization ability of SA3D. The evaluation results on the NVOS dataset is shown in Table 8.

## 5 Discussion

On top of the experimental results, we hope to deliver some insights from our preliminary study of integrating SAM and NeRF, *i.e.*, a 2D foundation model and a 3D representation model.

Table 6: Ablation on different IoU-aware rejection threshold $\tau$ on the Replica office0.

| $\tau$ | 0.3 | 0.4 | 0.5 | 0.6 | 0.7 |
|---|---|---|---|---|---|
| mIoU | 74.9 | 79.7 | 84.4 | 81.3 | 82.0 |

Table 7: Ablation on different negative term coefficient $\lambda$ on the Replica office0.

| $\lambda$ | 0.05 | 0.1 | 0.15 | 0.3 | 0.5 |
|---|---|---|---|---|---|
| mIoU | 79.1 | 82.9 | 84.4 | 84.9 | 83.3 |

Table 8: Ablation on different 2D segmentation models.

| SEEM [75] | | SimpleClick [32] | | RITM [49] | | FocalClick [5] | |
|---|---|---|---|---|---|---|---|
| mIoU (%) | mAcc (%) | mIoU (%) | mAcc (%) | mIoU (%) | mAcc (%) | mIoU (%) | mAcc (%) |
| 86.0 | 97.0 | 87.7 | 97.8 | 81.2 | 96.3 | 88.9 | 98.1 |

Input image with prompts 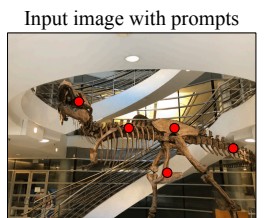 3 self-prompting points 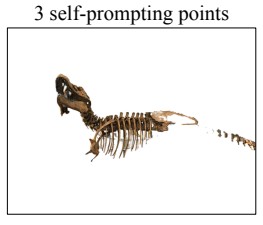 6 self-prompting points 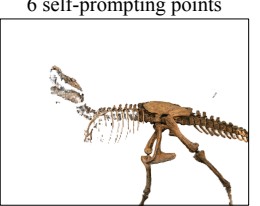 10 self-prompting points 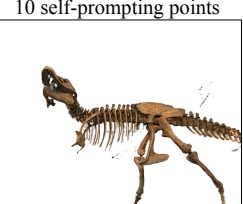

Figure 5: Results of different self-prompting points numbers $n_p$ on the LLFF-trex [37] scene.

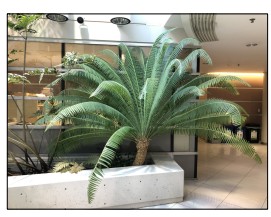 SAM prediction 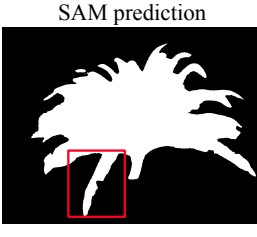 SA3D prediction 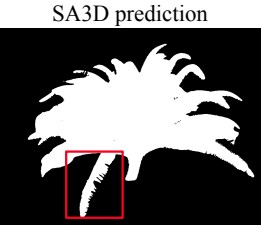

Figure 6: The 2D segmentation result by SAM and the 3D segmentation result by SA3D of the LLFF-fern [37] scene. SA3D produces more details that are missing in the SAM segmentation result.

First, NeRF improves the segmentation quality of SAM. In Figure 6, we show that SA3D can eliminate segmentation errors of SAM and effectively capture details such as holes and edges. Perceptually, SAM, as well as other 2D perception models, is often sensitive to the viewpoint, and NeRF offers the ability of 3D modeling and hence complementariness to assist recognition. Additionally, SA3D inspires us that using NeRF or other 3D structural priors is a resource-efficient method to lift a vision foundation model from 2D to 3D, as long as the foundation model has the ability to self-prompt. This methodology can save many resources because collecting a large corpus of 3D data is often costly. We look forward to research efforts to enhance the 3D perception ability of 2D foundation models (*e.g.*, injecting a 3D-aware loss into 2D pre-training).

**Limitation** SA3D has limitations in panoptic segmentation. First, the current paradigm relies on the first-view prompt. If some objects do not appear in the view for prompting, they will be omitted in the subsequent segmentation process. Second, the same part in the scene may be segmented into different instances with similar semantics in different views. This ambiguity cannot be easily eliminated under the current mechanism design and lead to unstable training. We leave these issues as future work.

## 6 Conclusion

In this paper, we propose SA3D, a novel framework that generalizes SAM to segment 3D objects with neural radiance fields (NeRFs) as the structural prior. Based on a trained NeRF and a set of prompts in a single view, SA3D performs an iterative procedure that involves rendering novel 2D views, self-prompting SAM for 2D segmentation, and projecting the segmentation back onto 3D mask grids. SA3D can be efficiently applied to a wide range of 3D segmentation tasks. Our research sheds light on a resource-efficient methodology that lifts vision foundation models from 2D to 3D.

## Acknowledgement

This work was supported by NSFC 62322604, NSFC 62176159, Natural Science Foundation of Shanghai 21ZR1432200, and Shanghai Municipal Science and Technology Major Project 2021SHZDZX0102. Specifically, We extend our sincere thanks to Weichao Qiu for his insightful suggestions during the course of this work.

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

## Appendix

The appendix includes the following contents:

1. Implementation details (Section A);
2. A segmentation refinement strategy (Section B);
3. The scribble to point strategy for the evaluation of NVOS (Section C);
4. An analysis about vanilla NeRF [38] used in SA3D (Section D);
5. More information about the object filtering of Replica [51] (Section E);
6. A further illustration of the self-prompting strategy (Section F);
7. An analysis about the effect of different kinds of occlusion in NeRF (Section G);
8. More visualization results with different kinds of input prompts (Section H).

## A    Implementation Details

We implement SA3D using PyTorch [42] with reference to the code provided by DVGOv2 [54]. The SA3D model is built and trained on a single Nvidia Geforce RTX3090 GPU. For our NeRF model, we primarily employ TensoRF [3], utilizing the VM-48 representation to store the radiance latent vectors. The radiance fields are pre-trained for most datasets with 40,000 iterations. For the LLFF dataset [37] and the 360 dataset [1], the radiance fields are trained with 20,000 iterations.

## B    Refinement with A Two-pass Segmentation Mechanism

SAM may produce segmentation masks containing undesired parts. The IoU-aware view rejection is hard to handle this issue when the mis-classified region gradually expands.

We propose a two-pass segmentation mechanism to further refine the segmentation result. After completing 3D segmentation introduced in the main manuscript, we get a 3D mask $\mathbf{V}$. To detect the mis-classified region from $\mathbf{V}$, we re-render the 2D segmentation mask $\mathbf{M}^u$ of the user-specific reference view and compare it with the original SAM segmentation result $\mathbf{M}^u_{\text{SAM}}$.

Subsequently, we reset the original 3D mask $\mathbf{V}$ to be a zero tensor and introduce another 3D mask $\mathbf{V}' \in \mathbb{R}^3$ that specifically indicates the mis-classified regions. The 3D segmentation process is then repeated, with the key difference being the incorporation of negative prompt points during the self-prompting phase. In other words, the prompts obtained from $\mathbf{V}'$ serve as negative prompts for $\mathbf{V}$, and vice versa. This incorporation of negative prompts enables SAM to gain a better understanding of the user's requirements and refine the segmentation accordingly (shown in Figure B7). It is important to note that while this two-pass segmentation mechanism holds promise, it was not utilized in our main experiments due to considerations of efficiency.

Vanilla SA3D                    Two-pass Segmentation

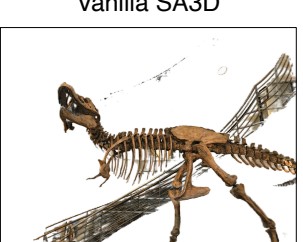 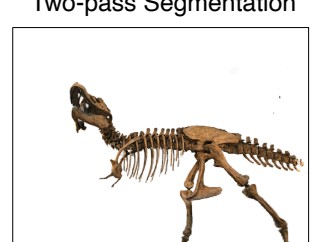

Figure B7: The effect of the two-pass segmentation refinement.

## C    The Scribble to Points Strategy for The Evaluation of NVOS

The NVOS [47] dataset provides a reference view and the corresponding scribbles for each scene (shown in Figure C8). In practice, since the scribbles usually contain tens of thousands of dense

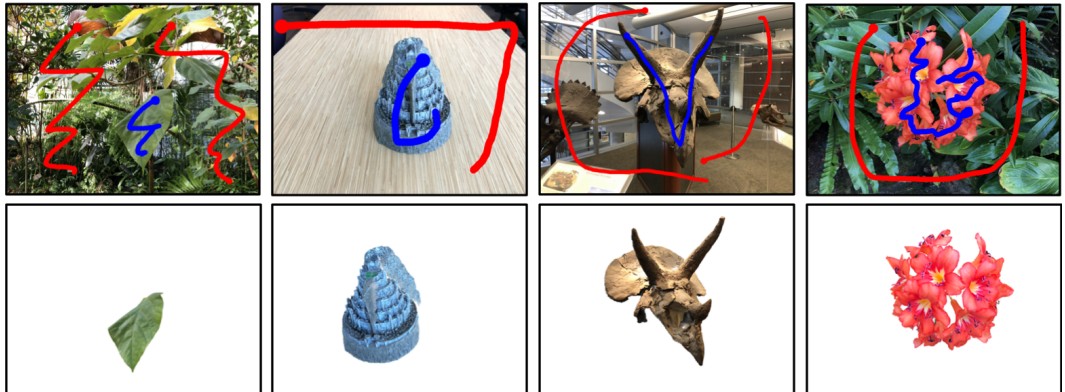

Figure C8: Some scribbles on the reference views provided by NVOS and the corresponding segmentation results of SA3D. Blue scribbles are positive and red scribbles are negative.

points, SAM [25] cannot directy take such scribbles as input. The abundance of points in the scribbles hinders SAM's performance when directly utilizing them as prompts, which is an inherent limitation of SAM.

For fair comparison, we extract positive and negative prompt points from the provided positive and negative scribbles, respectively. For input scribbles, we first skeletonize them and then select $2\%$ points from the skeletonized positive scribbles as the positive prompts and $0.5\%$ points from the skeletonized negative scribbles as the negative prompts.

## D    The Effect of Different NeRFs Used in SA3D

We adapt SA3D to the vanilla NeRF [38] to showcase its generalizability. We present visualization results on the LLFF dataset. As illustrated in Figure D9, SA3D with the vanilla NeRF exhibits excellent performance without the need for additional modifications.

## E    Object Filtering for The Replica Dataset

The Replica dataset contains many objects in each scene. However, it is important to note that many of these objects exhibit low quality, as depicted in Figure E10, making them unsuitable for evaluating 3D segmentation. Generally, these instances exhibit the following issues: some instances are not present in the training frames provided by Zhi *et al.* [73]; some instances are too small to be effectively segmented, such as thin slits in doors; and some instances consist of unrecognizable, low-resolution pixels, such as blurred tags, which are not suitable for accurate instance segmentation. Accordingly, we carefully select approximately 20 representative instances from each scene for the evaluation. The list of instance IDs for each scene can be found in Table E10. We have also included the quantitative results without object filtering in Table E9. Even without object filtering, SA3D demonstrates improvements compared to the single-view baseline.

Table E9: Quantitative results on Replica (mIoU) without object filtering.

| Scenes | office0 | office1 | office2 | office3 | office4 | room0 | room1 | room2 | mean |
|---|---|---|---|---|---|---|---|---|---|
| Single view | 58.8 | 53.1 | 61.2 | 54.2 | 56.7 | 51.0 | 58.6 | 58.3 | 56.49 |
| SA3D (ours) | **65.1** | **59.8** | **69.7** | **61.4** | **63.8** | **56.8** | **68.4** | **72.2** | **64.65** |

## F    An Illustration for The Proposed Self-prompting Strategy

We offer an illustration (Figure F11) to assist readers in gaining a clearer understanding of the self-prompting strategy.

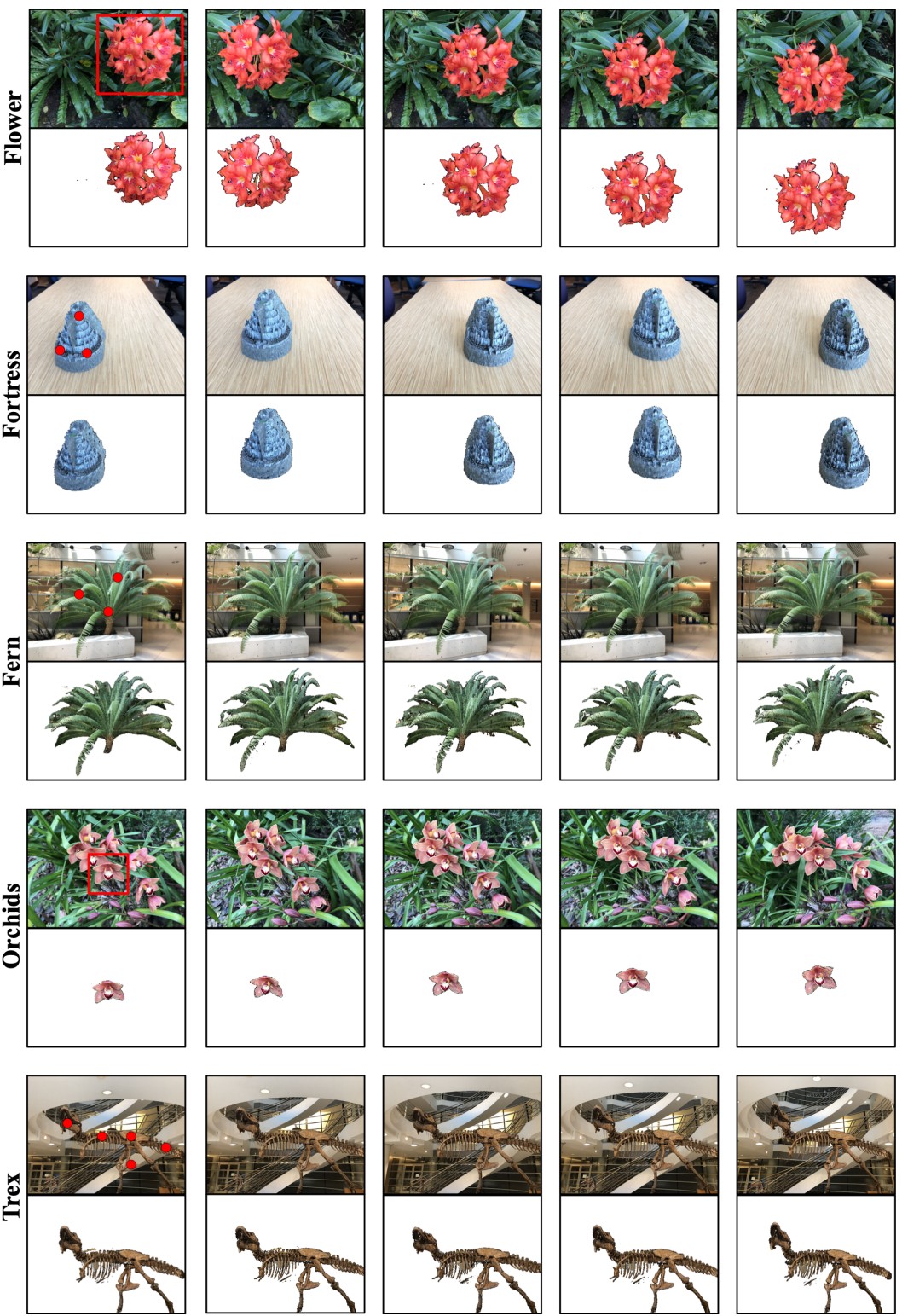

Figure D9: 3D Segmentation results based on the **vanilla NeRF** on the LLFF dataset.

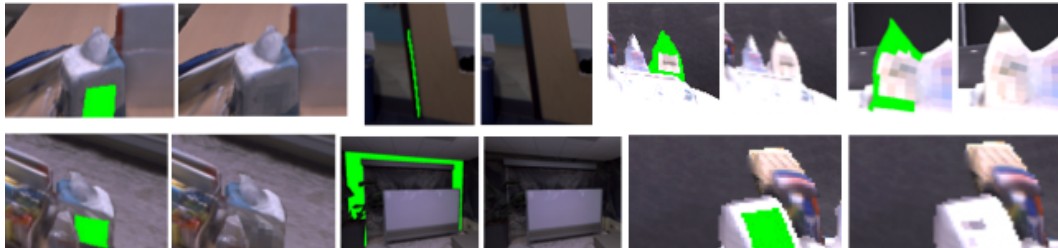

Figure E10: Some ground-truth masks (shown in green) and their corresponding instances from the Replica dataset. These unreasonable segmentation targets are filtered out in evaluation.

Table E10: The selected id lists of Replica.

| Scenes | ID list |
|--------|---------|
| office0 | 3,4,7,8,9,10,12,14,17,19,21,23,26,28,29,30,36,37,40,42,44,46,54,55,57,58,61 |
| office1 | 3,7,9,11,13,14,15,17,23,24,29,32,33,36,37,39,42,44,45,46 |
| office2 | 0,2,8,9,13,14,17,19,23,27,40,41,47,49,51,54,58,60,65,67,70,71,72,73,78,85,90,92,93 |
| office3 | 3,8,11,14,15,18,19,25,29,30,32,33,38,39,43,51,54,55,61,65,72, 76,78,82,87,91,95,96, 101,111 |
| office4 | 1,2,6,7,9,11,17,22,23,26,33,34,39,47,49,51,52,53,55,56 |
| room0 | 5,6,7,10,13,14,16,25,32,33,35,46,51,53,55,60,64,67,68,83,86,87,92 |
| room1 | 1,2,4,6,7,9,10,11,16,18,24,28,32,36,37,44,48,52,54,56 |
| room2 | 3,5,6,7,8,9,11,12,16,18,22,26,27,37,38,39,40,43,49,55,56 |

In the self-prompting strategy, prompt points $\mathcal{P}_s$ are derived from an incomplete 2D rendered mask $\mathbf{M}^{(n)}$, which is represented as a confidence score map. Initially, the selected prompt points set $\mathcal{P}_s$ is empty, and the first prompt point $\mathbf{p}_0$ is selected as the one with the highest confidence score in the mask $\mathbf{M}^{(n)}$. For subsequent points, square regions centered around existing prompt points are masked out on $\mathbf{M}^{(n)}$. The depth $z(\mathbf{p})$, estimated by the pre-trained NeRF, helps convert 2D pixel $\mathbf{p}$ into a 3D point $\mathcal{G}(\mathbf{p})$. The new prompt point is expected to have a high confidence score while being close to existing prompt points. Hence, a distance-aware decay term is introduced to compute the confidence score. The remaining point with the highest decayed score is added to the prompt set. This selection process is repeated until either the number of prompts $|\mathcal{P}_s|$ reaches a predefined threshold $n_p$ or the maximum value of the remaining points is less than 0. Please refer to Section 3.4 of the main manuscript for more details.

## G   The Effect of Occlusion in NeRF

There are two cases of occlusion in NeRFs: part of the target object does not appear in some views (but appear in other views), or it does not appear at all. For the former case, SA3D can recover it using information from other views; for the latter case, there can be parts missing in the target object (see Figure G12). This is an interesting future direction (*e.g.*, applying generative models such as diffusion models).

## H   More Visualization Results

We present additional visualization results in Figure H13 and Figure H15, showcasing the effectiveness of SA3D across various input prompts. We also provide some visualization of extracted meshes of the segmented objects (Figure H14) to show the extracted 3D geometry. Please note that the quality of these meshes can be further improved by applying more effective NeRF2Mesh methods [56, 66].

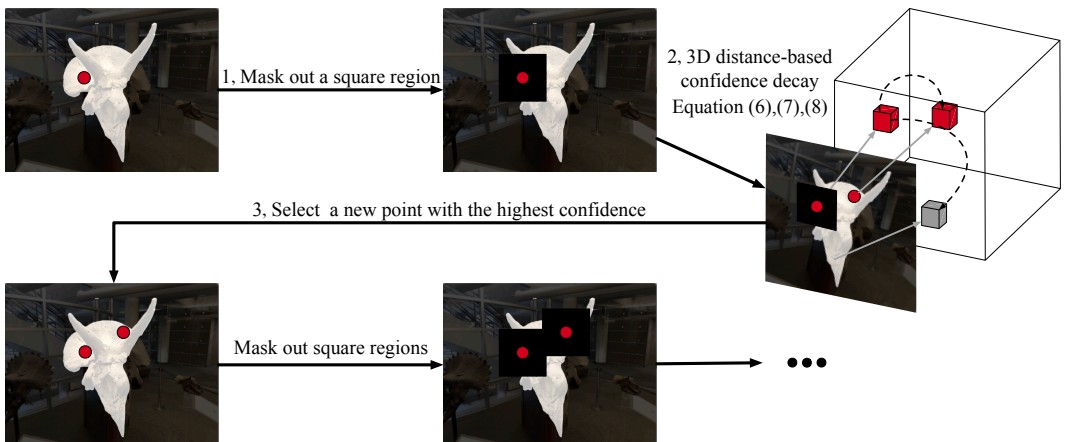

Figure F11: An illustration of the self-prompting strategy.

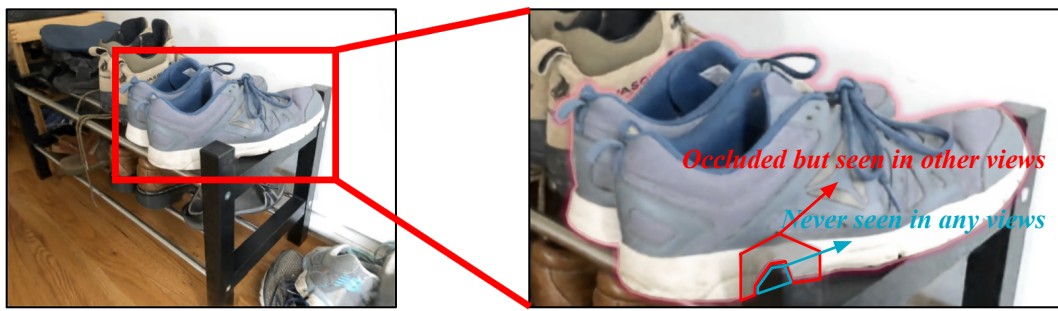

Figure G12: The effect of different kinds of occlusion in NeRF.

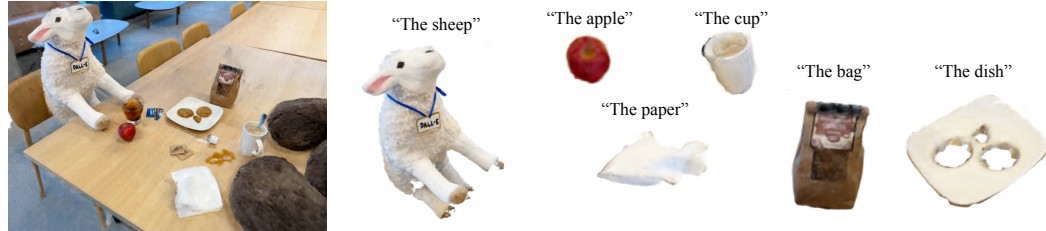

Figure H13: Text prompt based visualization results on the LERF figurines dataset.

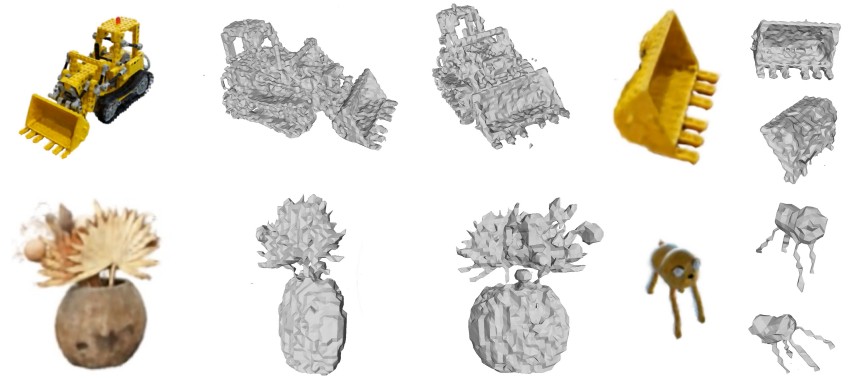

Figure H14: Some visualization results of the extracted meshes of the segmented objects.

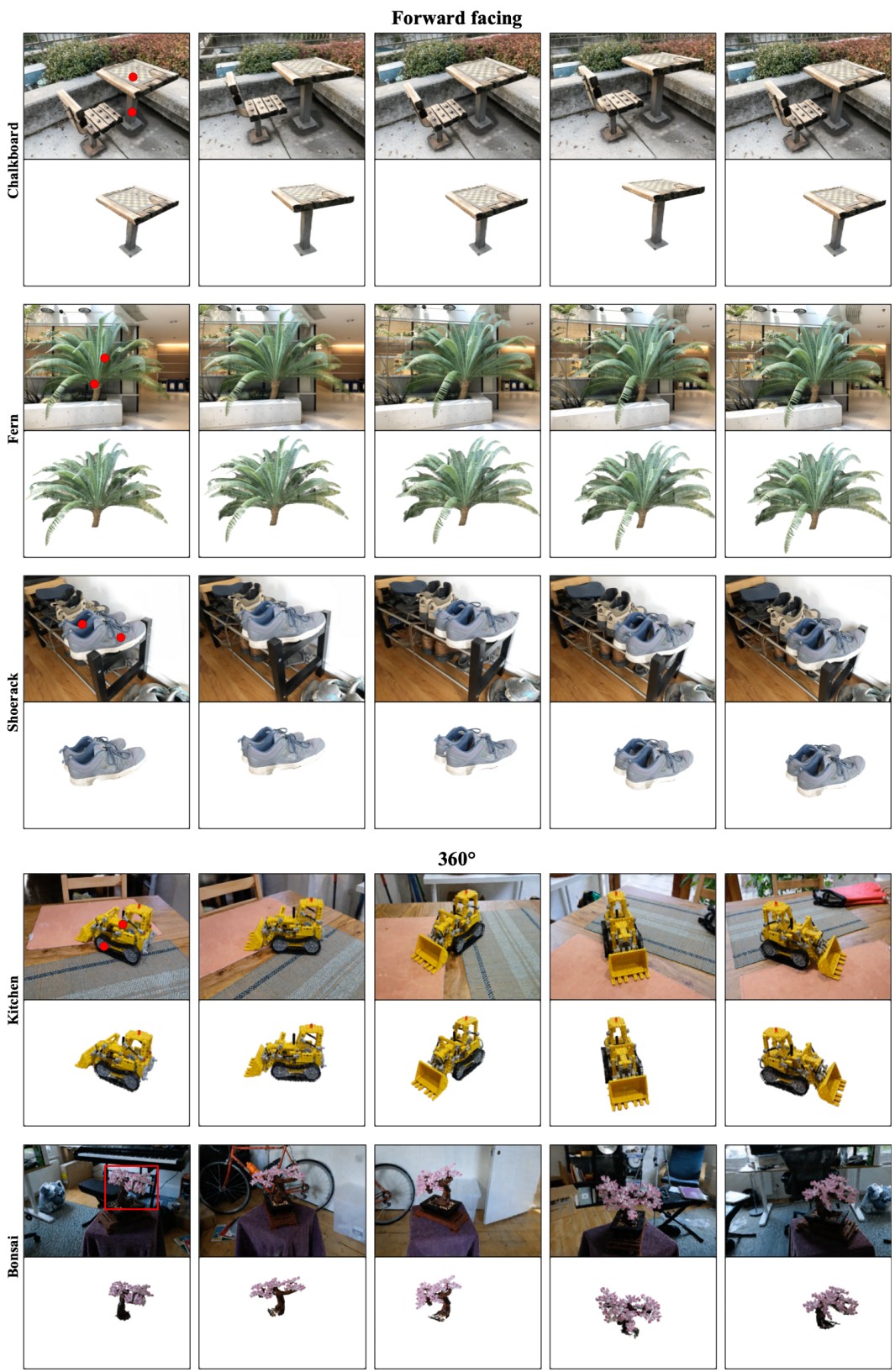

Figure H15: More visualization results on the LLFF dataset and the 360 dataset (based on point and box prompts).

