# OpenReview forum: "Segment Anything in 3D with NeRFs"
_NeurIPS.cc/2023/Conference — NeurIPS 2023 poster_

### Official Review · Reviewer_jjUN · 2023-07-04

**Soundness:** 3 good
**Presentation:** 4 excellent
**Contribution:** 3 good
**Rating:** 5
**Confidence:** 4

**Summary:**

This paper introduces a simple and efficient method for general 3D segmentation based on SAM (a powerful 2D foundation model) and NeRF-style representation. Instead of building a 3D foundation model from scratch, SA3D uses two steps to lifts 2D SAM segmentation results to 3D in a more concise and efficient way. Based on a well-trained NeRF, a rendered reference view and the human input prompts are sent into SAM to get the first segmentation. Then the proposed mask inverse rendering and cross-view self-prompting strategy will help optimize a volume-based 3D mask field and propagate segmentation information to different views in an iterative and incremental manner. The comprehensive experiments prove the effectiveness of the designed pipeline.

**Strengths:**

1.	The proposed method is simple but effective. It provides a general interactive 3D object segmentation paradigm which do not rely on heavy pre-training.
2.	The method is pretty efficient. Given a pre-trained NeRF, the 3D segmentation can be completed within only minutes with the help of SAM.
3.	The experiments have covered variant datasets and comparison to SOTA methods, which are comprehensive and persuasive.
4.	The paper is well-written and easy to follow.


**Weaknesses:**

1.	I expect that segment “anything” in 3D means the method can segment all the things in a 3D scene, so that it is consistent with the purpose of SAM. But the proposed SA3D can only segment one object at each time.
2.	Furthermore, using prompts (such as points, scribbles, text) to achieve 2D segmentation of one target object is a long-studied subject. Besides SAM, there should be many alternative choices such as [1][2][3], which should have been evaluated for this 3D task. Or the irreplaceability of SAM needs to be explained.
3.	I wondering if the method requires complete observation of the target object in at least one image. What if the target object is not observed completely in any views? How will the choice of reference view affect the 3D segment results? Is there any necessary strategy to achieve best performance? The robustness of the method should be evaluated.

[1] Sofiiuk K, Petrov I A, Konushin A. Reviving iterative training with mask guidance for interactive segmentation[C]//2022 IEEE International Conference on Image Processing (ICIP). IEEE, 2022: 3141-3145.
[2] Liu Q, Xu Z, Bertasius G, et al. SimpleClick: Interactive image segmentation with simple vision transformers[J]. arXiv preprint arXiv:2210.11006, 2022.
[3] Chen X, Zhao Z, Zhang Y, et al. Focalclick: Towards practical interactive image segmentation[C]//Proceedings of the IEEE/CVF Conference on Computer Vision and Pattern Recognition. 2022: 1300-1309.



**Questions:**

1.	It seems that the method is not restricted by NeRF-based representation. Since the authors assume NeRF provides good depth, the proposed method should also work under other representations such as mesh and point cloud?
2.	The 3D mask is modeled as dense voxel grid, which may cause high memory consumption for high resolution representation. How will different resolutions affect the performance? Maybe using a MLP is a better choice?


**Limitations:**

Yes.

---

> ### Author Rebuttal · Authors · 2023-08-10
>
> We sincerely thank you for your valuable comments. We answer your questions as follows and hope the response could clear your concerns.
>
> ### Weaknesses
>
> > W1: ... SA3D can only segment one object at a time.
>
> **A1:** SAM achieves “segment everything” by densely prompting on the image. These prompts are stacked at the batch dimension and fed to the model. Such implementation, in essence, is equivalent to querying the SAM decoder for many objects one by one.
>
> SA3D also inherits the ability and mechanism. Segmenting multiple objects can be achieved by stacking multiple 3D mask grids into a 4D data structure (i.e. HxWxL -> NxHxWxL), where N denotes the number of objects. Then, iterative self-prompting and inverse rendering can be performed for all objects simultaneously. We attach some visualization results in Fig 3 (global response PDF).
>
> By the way, as stated in Lines 282-285, we admit that the current method still has limitations in segmenting everything. For example, it is difficult for SAM to guarantee cross-view consistency for extremely small parts, because some parts may be segmented into different instances with similar semantics in different views. Fixing these cases in 3D is a challenging issue and is left for future research.
>
> > W2: ... Besides SAM, alternative choices [1-3] ...
>
> **A2:** Yes. SA3D can generalize to these methods effortlessly, which demonstrates our claim in the abstract: "... lift a 2D vision foundation model to 3D, as long as the 2D model can steadily address promptable segmentation across multiple views". We conduct experiments on the NVOS dataset and the 'Replica office 0' scene to evaluate SA3D with these interactive segmentation methods:
>
> ||SAM||SimpleClick||RITM||FocalClick||
> |:-:|-|-|-|-|-|-|-|-|
> |Metrics|IoU|Acc|IoU|Acc|IoU|Acc|IoU|Acc|
> |Mean|90.3|98.2|87.7|0.9778|81.2|96.3|88.9|98.1|
>
> |Replica_office_0|SAM|SimpleClick|RITM|FocalClick|
> |-|-|-|-|-|
> |mIoU|84.4|72.7|69.0|63.6|
>
> From the above results we find lifting these interactive segmentation methods with SA3D can beat the previous SOTA (ISRF, mIoU 83.8; NVOS, mIoU 70.1) on the NVOS benchmark. Additionally, when encountered with complicated indoor scenes like Replica, other methods perform much worse, showing the robustness of SA3D as a foundation model.
>
> > W3: What if not observed completely in any views? ... the choice of reference view ...? ... necessary strategy ...? robustness ...
>
> **A3:** There are two cases: part of the object does not appear in some views (but appear in other views), or it does not appear at all. For the former case, SA3D can recover it using information from other views; for the latter case, there can be parts missing in the target object (see Fig 9, global response PDF). This is an interesting future direction (e.g. applying generative models such as diffusion models).
>
> To evaluate the robustness of SA3D, we randomly select 3 new reference views from the training set and compute the mean metrics on the NVOS dataset. The results are as follows:
>
> ||Random 3 views||Reported in the Paper||
> |-|-|-|-|-|
> |Metrics|IoU|Acc|IoU|Acc|
> |Mean|89.8|98.2|90.3|98.2|
>
> Visualization results are provided to demonstrate the robustness of SA3D to reference view. Please check Fig 8 (global response PDF).
>
> ### Questions
>
> > Q1: ... not restricted by NeRF ... mesh and point cloud?
>
> **A4:** Yes. Extending SA3D to other 3D formats is straightforward. Here we elaborate a pipeline for point clouds:
> 1. Generate multi-view images on the point cloud data using off-the-shelf methods (e.g. Point2Pix, CVPR'23). Practically, we find projecting 3D points with RGB information onto a 2D plane is enough for SAM to segment
> 2. Select a reference view and input prompts; use SAM for segmentation
> 3. Assign the predicted mask to the corresponding 3D points (Mask Inverse Rendering)
> 4. Project the 3D mask onto a new view and get the corresponding 2D image (Mask Rendering)
> 5. Extract self-prompting points from the 2D mask (Self-prompting)
> 6. Feed the prompt and 2D image to SAM for segmentation
> 7. Repeat 3-6 until getting the full object
>
> > Q2: ... dense voxel grid ... high memory consumption ... different resolutions affect ... MLP is better?
>
> **A5:** Good question! In our implementation, the resolution of the mask grids is set to be TensoRF grids (320^3). We study different resolutions of mask grids on the NVOS dataset, results shown below and visualization provided in Fig 7 (global response PDF). The effect of the grid resolution is slight.
>
> |$320^3$||$160^3$||$80^3$||$40^3$||MLP||
> |-|-|-|-|-|-|-|-|-|-|
> |mIoU|mAcc|mIoU|mAcc|mIoU|mAcc|mIoU|mAcc|mIoU|mAcc|
> |90.3|98.2|90.1|98.2|89.2|98.1|88.0|97.8|80.9|96.4|
>
> We also report the results of an MLP version (not better) in the above table. Below are the advantages of explicit mask grids over MLP.
>
> 1. Optimization for mask grids is explicit and straightforward. In contrast, assigning a 3D point as positive in an MLP may unexpectedly make other points positive in the 3D space, leading to unstable self-prompting.
> 2. In gradient descent, using explicit mask grids saves a lot of memory because its computation graph is simple. For MLP, the computation graph contains too many non-leaf nodes for different layers of the MLP.
> 3. By adopting mask grids, the 3D segmentation results are explicit, which makes the downstream task, *e.g.* editing, more convenient. For example, removing the target object or extracting it from the scene only requires direct matrix multiplication between the mask grids and the density grids (if the density grids are also stored explicitly).
>
> We agree that MLP enjoys theoretically infinite resolution and smaller storage costs. Therefore, exploring hybrid mask representations is a promising future direction.

---

> > ### Author Response · Authors · 2023-08-13
> > **Fixing Two Typos in the Rebuttal**
> >
> > Dear reviewer,
> >
> > We would like to fix two typos in our rebuttal. In **A2**, the Acc of SimpleClick (in the table) should be **97.8** rather than 0.9778. Additionally, the final sentence "… showing the robustness of SA3D as a foundation model." should be revised to "… showing the robustness of **SAM** as a foundation model."
> > We apologize for any potential misunderstanding that may have arisen.
> >
> > Best,
> >
> > Authors

---

### Official Review · Reviewer_b3JM · 2023-07-06

**Soundness:** 3 good
**Presentation:** 4 excellent
**Contribution:** 3 good
**Rating:** 7
**Confidence:** 5

**Summary:**

This paper proposes to lift 2D segmentations from foundation models such as SAM to 3D by iterating between SAM and NeRF, without re-training or re-defining either. Given a trained NeRF model, the model first renders a view, which is also processed by SAM given a user click. With the segmentation by SAM, the model optimizes a 3D segmentation volume such that it volume renders into a mask consistent with what SAM produced (“mask inverse rendering”). Next, the model projects this initial segmentation volume to other viewpoints, producing incomplete 2D masks. Finally, the model computes “good prompts” for SAM to complete these masks (“cross-view self prompting”). The model iterates between these steps until a complete segmentation volume has been produced.

Because of the generalization power of SAM, the model is able to segment almost anything in 3D, without requiring applying changes to SAM or NeRF, making itself a framework that can be applied to any 2D foundation models that we want to lift to 3D.


**Strengths:**


The proposed method is simple yet effective, following the recent trend of developing foundation models and/or open-vocabulary LLMs. More importantly, it also bridges the gap between powerful 2D models and 3D understanding as required by robots or autonomous vehicles. It is general and applicable to any 2D foundation models. The results are strong both qualitatively and quantitatively.



**Weaknesses:**

Since the framework is claimed to be (and I think it is) general and applicable to any 2D foundation models, the paper will be much stronger if the authors could demonstrate the use of this framework to lift another foundation model’s output into 3D.

The paper will also benefit from showing some 3D shape results that are extracted from the 3D segmentation volume. Often, it’s the underlying 3D geometry that matters for, say, robotic manipulation. “RGB looking good” is separated than that.


**Questions:**


Has runtime been reported? I recommend augmenting Table 4 with an additional row of runtime.




**Limitations:**

Yes

---

> ### Author Rebuttal · Authors · 2023-08-09
>
> We sincerely thank you for your valuable feedback and hope our following clarifications and responses could clear your concerns.
> ### Weaknesses
>
> > W1: Demonstrate the use of this framework to lift another foundation model’s output into 3D.
>
> **A1:** Thanks for the suggestion. We try to lift SEEM [1] with SA3D, which is a concurrency work of SAM. As suggested by Reviewer jjUN, we also lift three interactive methods with SA3D. SEEM supports more modals but suffers from relatively worse segmentation performance. We evaluate the performance of SA3D with SEEM (and the three other interactive methods) on the NVOS dataset. The results are shown as below:
>
> |              | SEEM       |        | SimpleClick |           | RITM      |           | FocalClick |           |
> |:------------:|------------|--------|-------------|-----------|-----------|-----------|------------|-----------|
> |     Scene    | IoU        | Acc    | IoU         | Acc       | IoU       | Acc       | IoU        | Acc       |
> |     fern     | 0.7921     | 0.9293 | 0.8109      | 0.9347    | 0.7662    | 0.9154    | 0.8214     | 0.9404    |
> |    flower    | 0.9313     | 0.9836 | 0.9362      | 0.9849    | 0.8912    | 0.9728    | 0.9372     | 0.9852    |
> |   fortress   | 0.9851     | 0.9972 | 0.9761      | 0.9954    | 0.9755    | 0.9953    | 0.9839     | 0.997     |
> | horns_center | 0.8182     | 0.9627 | 0.9554      | 0.9921    | 0.8642    | 0.9759    | 0.9547     | 0.9920     |
> |  horns_left  | -          | -      | 0.8599      | 0.9908    | 0.8451    | 0.9896    | 0.8407     | 0.9894    |
> |    leaves    | -          | -      | 0.9326      | 0.9957    | 0.9341    | 0.9958    | 0.9258     | 0.9952    |
> |    orchids   | 0.8655     | 0.9771 | 0.7869      | 0.9636    | 0.5744    | 0.9089    | 0.8984     | 0.9827    |
> |     trex     | 0.7700       | 0.9701 | 0.7598      | 0.9647    | 0.6418    | 0.9472    | 0.7507     | 0.9636    |
> |     mean     | 0.8604 | 0.9700   | 0.8772    | 0.9778 | 0.8116 | 0.9626 | 0.8891     | 0.9807 |
>
> Please kindly note that the missing entries for SEEM are because it cannot generate reasonable segmentation results for the reference view images no matter how we adjust the prompts. Some visualization results of SEEM are provided in Fig. 2 of the global author response PDF. SEEM has the advantage of accommodating multiple cross-modal prompt inputs, but exhibits inferior segmentation performance compared to SAM. Further utilizing the cross-modal prompting ability of different foundation models to enhance the behaviour of self-prompting is a promising direction.
>
> > W2: Showing some 3D shape results that are extracted from the 3D segmentation volume.
>
> **A2:** We extract meshes from the segmented 3D objects based on the standard marching cubes algorithm. Visualization results can be found in Figure 4 of the global author response PDF. Please note thatthe quality of these meshes can be further improved by applying more effective NeRF2Mesh methods [2][3]. The scripts for mesh extraction will also be released to facilitate the following research.
>
> ### Questions
>
> > Q1: Augmenting Table 4 with an additional row of runtime.
>
> **A3:** We update Table 4 with runtime supplemented as follows. It shows a trade-off between time cost (related to the number of views) and the segmentation quality.
>
> | |||||
> |----------------------------------|----------|----------|----------|------------|
> | Number of Views                  | 5 (10%)  | 9 (20%)  | 21 (50%) | 43 (100%)  |
> | IoU on Fortress (forward facing) | 97.8     | 98.3     | 98.3     | 98.3       |
> | Time Cost(s)                       | 7.56    | 12.80   | 28.98   | 58.97     |
> | Number of Views                  | 11 (10%) | 21 (20%) | 51 (50%) | 103 (100%) |
> | IoU on Lego (360 degrees)        | 84.5     | 84.8     | 91.5     | 92.2       |
> | Time Cost(s)                        | 23.49    | 43.54    | 103.83   | 204.93     |
>
>
> [1] Zou, Xueyan, et al. "Segment everything everywhere all at once." arXiv preprint arXiv:2304.06718 (2023).
>
> [2] Tang, Jiaxiang, et al. "Delicate textured mesh recovery from nerf via adaptive surface refinement." arXiv preprint arXiv:2303.02091 (2023).
>
> [3] Yariv, Lior, et al. "BakedSDF: Meshing Neural SDFs for Real-Time View Synthesis." arXiv preprint arXiv:2302.14859 (2023).

---

> > ### Comment · Reviewer_b3JM · 2023-08-13
> > **Post-Rebuttal Update**
> >
> > I thank the authors for an informative rebuttal, which includes several interesting new results. Provided that the authors will include these new results -- lifting another foundation model, 3D mesh extraction, etc. -- in their final paper (which will be a much stronger one), I remain positive about this paper and support its acceptance.

---

> > > ### Author Response · Authors · 2023-08-13
> > > **Thanks for Your Response**
> > >
> > > Thank you very much for your response and efforts! Your comments have greatly improved our paper. The related new content will be incorporated into our final version.

---

### Official Review · Reviewer_JhYG · 2023-07-09

**Soundness:** 3 good
**Presentation:** 3 good
**Contribution:** 3 good
**Rating:** 5
**Confidence:** 5

**Summary:**

This paper proposes a method for segmenting a pre-trained NeRF by utilizing SAM. Given a pre-trained NeRF, it first asks users to provide prompts (e.g., some points) for a reference view. It then utilizes the SAM to generate a 2D segmentation for the reference view and utilizes the 2D mask to optimize a 3D segmentation mask. After that, it will iteratively update the 3D mask by (a) selecting a random review, (b) rendering the view, (c) utilizing the 3D mask to automatically generate prompts for the view, (d) utilizing prompts to query SAM, (e) utilizing SAM output to update 3D mask. The method proposes a self-promoting strategy to generate prompts given the 3D mask and an IoU-aware view rejection to ignore the bad prediction from SAM.

**Strengths:**

1. The paper proposes a novel method for segmenting a pre-trained NeRF by using SAM.
2. The paper proposes a self-promoting strategy for automatically generating prompts for SAM and a rejection strategy to ignore bad SAM predictions.
3. The authors conduct experiments on three datasets and provide some ablation studies.
4. The paper is easy to follow.

**Weaknesses:**

1. The method requires minutes for a single segmentation (e.g., of an object), which may greatly limit the usage of the method in many real-world applications (e.g., robotics manipulation).

2. Recently, there are also many NeRF/Point Cloud-based open-world (vocabulary) 3D segmentation (for both scene-level and part-level) methods[1-8] that leverage pre-trained 2D VLM (e.g., CLIP). However, the discussion and comparison with them is missing. It seems that many of these prior methods don't need per-instance optimization and can generate a 3D segmentation mask in just seconds. Please cite these papers and discuss the advantages of the proposed methods.

3. In Line 153, the paper states, "Given an incomplete 2D rendered mask". Why is the rendered mask always incomplete? Is it possible that some SAM predictions are wrong and include extra regions, which leads to an enlarged 3D segmentation mask? If this is possible, the self-prompting strategy will also generate a wrong prompt for SAM in the later steps. Please explain whether this case is possible and how the proposed method can handle the wrong SAM prediction (including extra regions).

4. The negative refinement term (Line 137) is unclear to me. Please explain in more detail about the motivation for this term.

5. Equation (7) is not clear to me. Could you explain in more detail? Also, it would be better to provide an ablation study to verify the necessity of the confidence decay step (Equation (7)). Can the self-prompting strategy still work without confidence decay?

6. For all tables, could you include runtime for both the proposed method and baseline methods?

7. It would be better to include evaluations on some standard 3D segmentation benchmarks (e.g., ScanNet and PartNet) to have extensive comparison with existing methods as well.



[1] Kerr, Justin, et al. "Lerf: Language embedded radiance fields." arXiv preprint arXiv:2303.09553 (2023).

[2] Peng, Songyou, et al. "Openscene: 3d scene understanding with open vocabularies." Proceedings of the IEEE/CVF Conference on Computer Vision and Pattern Recognition. 2023.

[3] Ding, Runyu, et al. "PLA: Language-Driven Open-Vocabulary 3D Scene Understanding." arXiv preprint arXiv:2211.16312 (2022).

[4] Ha, Huy, and Shuran Song. "Semantic abstraction: Open-world 3d scene understanding from 2d vision-language models." 6th
Annual Conference on Robot Learning. 2022.

[5] Zhang, Junbo, Runpei Dong, and Kaisheng Ma. "Clip-fo3d: Learning free open-world 3d scene representations from 2d dense clip." arXiv preprint arXiv:2303.04748 (2023).

[6] Liu, Minghua, et al. "Partslip: Low-shot part segmentation for 3d point clouds via pretrained image-language models." Proceedings of the IEEE/CVF Conference on Computer Vision and Pattern Recognition. 2023.

[7] Yang, Jihan, et al. "Regionplc: Regional point-language contrastive learning for open-world 3d scene understanding." arXiv preprint arXiv:2304.00962 (2023).

[8] Jatavallabhula, Krishna Murthy, et al. "Conceptfusion: Open-set multimodal 3d mapping." arXiv preprint arXiv:2302.07241 (2023).


**Questions:**

1. How many gradient descent iterations do you need in each inverse rendering step?

2. In Table 2, SA3D performs worse than the baseline method (MVSeg). Is there any common pattern for these instances? Or any reasons for the performance differences?


**Limitations:**

Yes

---

> ### Author Rebuttal · Authors · 2023-08-10
>
> Thanks for your instructive comments.
>
> ### Weaknesses
> > W1: ... minutes for a single segmentation ...
>
> **A1:** The time cost reported in our paper is an upper bound. As shown in Table 4 (global response text), SA3D achieves satisfactory segmentation with a few sampled views, which only requires < 10 seconds.
>
> > W2: ... [1-8] that leverage VLM ...  cite and discuss ...
>
> **A2:** Thanks. We will cite all these papers and add discussions below.
>
> The most relevant work to SA3D is LERF [1], which trains a feature field of the VLM together with the radiance field. Compared with SA3D, LERF focuses on coarsely localizing the specific objects with text prompts but not fine-grained 3D segmentation. The reliance on CLIP features makes it insensitive to the specific location information of the target object. When there are multiple objects with similar semantics in the scene, LERF cannot perform effective 3D segmentation. We evaluate the segmentation ability of LERF on the NVOS dataset and attach the results in the global response PDF. Moreover, we also provide the visualization results in Fig 6 of the global response PDF to support our statement.
>
> The remaining methods mainly focus on point clouds. By connecting the 3D point cloud with specific camera poses with 2D multi-view images, the extracted features by VLM can be projected to the 3D point cloud. The data acquisition of these methods is more expensive than ours, i.e., acquiring multi-view images for NeRFs.
>
> > W3: Line 153 ... rendered mask always incomplete? ... handle the wrong SAM prediction？
>
> **A3:** Sorry for the ambiguous statement "incomplete 2D rendered mask". We will replace "incomplete" with "inaccurate".
>
> It is possible that SAM generates inaccurate predictions. We designed several mechanisms to tackle this problem:
> - The negative refinement term in the mask inverse rendering loss (Eq. 5). In each iteration, if a region is predicted as background by SAM, its mask confidence score is then suppressed, which significantly alleviates inaccurate segmentation by SAM under certain views.
> - The confidence decay term in the self-prompting strategy (Eq. 8). This term adjusts rendered mask confidence scores according to the distance in the 3D space between the selected prompts and the candidate coordinates. This facilitates closely prompted points in 3D.
> - The IoU-aware view rejection (Lines 171-176). When SAM predicts a mask that greatly differs from the currently rendered one, i.e., obtaining a low IoU, this view will be skipped to avoid wrong mask allocation.
>
> > W4: The negative refinement term ... unclear ...
>
> **A4:** This term is used to suppress the extra regions included in SAM predictions on the mask grids: When a region is not segmented as foreground by SAM, its mask confidence score is suppressed. Thus the mask grids are labeled as foreground only if the region is consistently classified as foreground in different views. We will clarify it in Lines 135-138.
>
> > W5: Eq. (7) not clear ... ablation study...?
>
> **A5:** Eq. 7 defines a confidence decay term for self-prompting, based on the intuition that prompt points should not be too far apart in the 3D space. Assuming we have gathered N prompt points. When we try to select the (N+1)-th point from the remaining candidates, we first check the distance between these candidates and the existing N prompt points. If a candidate is far from all of them, the confidence score is suppressed heavily.
>
> To realize this, for a candidate, we traverse the existing prompts and get a set of decay terms. The smallest of them is determined as the final decay term for the candidate. This decay term between a candidate and an existing prompt point is defined as a product involving the confidence score of the prompt point and the min-max normalized 3D Euclidean distance between the two points.
>
> In common cases, without the decay step, the self-prompting still works. But for some hard cases, it may fail. For the concrete ablation and discussion, please refer to our response to R-Xgwo's Q2.
>
> > W6: ... runtime ...
>
> **A6:** See the global response texts for the time cost.
>
> > W7: ... evaluations on some standard 3D segmentation benchmarks ...
>
> **A7:** Please kindly note SA3D performs interactive segmentation, which is quite different from traditional 3D segmentation approaches. However, making SA3D support traditional 3D segmentation is an interesting research direction. We still provide the experimental results on Scannet for a more comprehensive evaluation. The experimental setting follows Table 3. Some results (mIoU) are shown as follows:
>
> ||scannet0050_02|scannet0144_01|scannet0300_01|scannet0354_00|scannet0389_00|
> |-|-|-|-|-|-|
> |Single View|53.0|63.5|61.2|56.9|60.6|
> |SA3D|72.9|77.1|75.8|69.8|78.9|
>
> ### Questions
>
> > Q1: ... iterations ... in each inverse rendering step?
>
> **A8:** Only one gradient descent iteration is required.
>
> > Q2: In Table 2, SA3D performs worse than ... MVSeg.
>
> **A9:** Yes. This gap stems from the inductive bias of SAM and some ambiguous segmentation targets. We provide some visualization results to support the statement in Fig 5 (global response PDF).
> In the 'Orchids' scene, the segmentation target is a group of flowers. SAM tends to segment each flower separately. If SAM is forced to segment them into one, it may involve some unexpected regions in the prediction.
> A similar phenomenon happens in the 'Room' scene. The SPIn-NeRF dataset treats the table along with objects on it as a whole. SAM segments the table separately, ignoring the placed objects.
> For the 'Pinecone' scene, both MVSeg and SA3D perform well. This scene involves many details, which are ignored by the ground-truth sometimes. We believe a 0.5% mIoU gap is reasonable since both our segmentation results and corresponding annotations are not perfect.

---

> > ### Comment · Reviewer_JhYG · 2023-08-20
> > **Thank you!**
> >
> > Thank you for your detailed response! Most of my concerns have been addressed.

---

> > > ### Author Response · Authors · 2023-08-21
> > > **Thanks for Your Response**
> > >
> > > We greatly appreciate your response and once again extend our sincere gratitude for your valuable time and effort spent on the review. If there are any points that require further clarification, we wholeheartedly welcome additional discussion.

---

### Official Review · Reviewer_Xgwo · 2023-07-11

**Soundness:** 3 good
**Presentation:** 3 good
**Contribution:** 3 good
**Rating:** 7
**Confidence:** 3

**Summary:**

The authors propose a combination of the newly introduced Segment-Anything Model (SAM) with Nerf, yielding the  Segment Anything in 3D (SA3D) system.
SA3D cam take a 3D scene reconstructed by Nerf and based on a user prompt (e.g. a few keypoints) can carve out distinct 3D objects from the scene. This is shown to outperform previous state-of-the-art systems.
They two novel bits to successfully combine the two approaches:
Firstly a loss function in order to induce a 3D mask field on a voxel grid based on the SAM outputs (Eq. 4) and secondly (and most importantly) a method to prompt SAM on another view based on the existing 3D mask view - allowing one to only prompt from one image and then perform "prompt propagation" to the rest.

**Strengths:**

Simple approach that should be easy to reproduce - the authors also share their code, which looks reasonably clean.

Solid experimental improvements - I am convinced by these numbers that the method outperforms some of the latest state-of-the-art systes.

Good ablations.

Interesting -but a bit hand-wavy - results that SA3D can improve SAM; some more validation of this would be welcome.

**Weaknesses:**

- Overstatement: the authors are  using SAM in tandem with Nerf, to get 3D objects out of a scene, which is great - but practically this is similar to classic co-segmentation or the systems that they compare to, but a bit better because of piggy-backing on SAM.
Stating in the abstract that  "Our research offers a generic and efficient methodology to lift a 2D vision foundation model to 3D, as long as the 2D model can steadily address  promptable segmentation across multiple views." suggests more than just this - one can start imagining extending the attention operations to 3D, supervising for depth/volumetric reconstruction etc, or more importantly having a foundation model for 3D, none of which is  the case based on what we have in the present paper. Technically the statement is not false - but implies more than what is actually happenning in the paper.


- I could not find any discussion about the computational efficiency of the proposed algorithm. At the moment it's unclear what is the importance of this.

Minor:

- Unclear intuitively what the second term does in Eq. 5 from the way it's written. I think a more obvious rewrite is
(lambda - 1) L_proj + lambda sum_r M(r) \propto L_proj + lambda/(lambda-1) sum_r M(r),
and explain that the second term is just a regularization term on the optimized segmentation field.

- l. 280: "we demonstrate limitations of SA3D in panoptic segmentation" -> could not find any pointer to this in the paper.

**Questions:**

-some more validation of the statement that SA3D can improve SAM would be welcome.

- Self prompting strategy: I am not too sure I see the necessity of this particular prompting strategy. The authors are lifting every prompt point to 3D, reducing the mask score in 3D in its neighborhood, regenerating the mask, and picking the next strongest point. How much better is this than a plain 2D-based nonmaximum suppression strategy? This should be  more efficient computationally and yield roughly the same results - if not, how much worse are they?

**Limitations:**

Yes

---

> ### Author Rebuttal · Authors · 2023-08-09
>
> We sincerely thank you for your efforts on the review work and your detailed comments. We answer your questions as follows and hope the response could clear your concerns.
> ### Weaknesses
> > W1: Overstatement: ... "Our research ... as long as ..." suggests more ...
>
> **A1:** Thanks for the suggestion. We acknowledge that SA3D may not have all the abilities you mentioned, but it indeed has some valuable potential. To better support our statement, we demonstrate the generalization ability of our framework with some other models (please check our **A1** to b3JM).
>
> Following your suggestion, we change our statement to “Our research reveals a **potential** methodology to lift **the ability** of a 2D vision foundation model to 3D, as long as the 2D model can steadily address promptable segmentation across multiple views.” We welcome any other suggestions on improving this part.
>
> > W2: ... discussion about the computational efficiency ...
>
> **A2:** Please check the global response for a comprehensive discussion about computation efficiency.
>
> > W3: Unclear ... Eq. 5 ... a more obvious rewrite ...
>
> **A3:** Thanks for the suggestion. The second term is essentially a regularization term that adds a negative effect on all regions involved in rendering the current mask to suppress the potential inaccurately-segmented region.
> Only if SAM consistently predicts a region as foreground from different views, SA3D marks its corresponding 3D region as foreground. We will clarify the statement around Line 135-138 and take this suggested equation as a supplement for better understanding.
>
> > W4: l. 280: "we demonstrate limitations ... " -> could not find ...
>
> **A4:** Sorry for the misleading. Here what we want to express is we admit that SA3D has some limitations in panoptic segmentation.
> We will revise the statement in Line 280 as “SA3D has limitations in panoptic segmentation”. Hope this can resolve the ambiguity.
>
> ### Questions
>
> > Q1: ... validation of ... SA3D can improve SAM ...
>
> **A5:** To further demonstrate our statement, we provide more visualization results (Fig. 1 of the global response PDF) on the 'bonsai' scene of the Mip-NeRF 360 dataset. Here we give a detailed explanation about why “SA3D can improve SAM” for clarification.
>
> Segmentation models often face challenges in accurately capturing object details such as small holes and gaps due to limitations in resolution. Even though SAM exhibits fine-grained segmentation capabilities, this issue still persists.
>
> The ability to improve SAM's performance stems from the fine-grained depth estimation (or the geometry information) provided by NeRF. The utilization of such information to aid segmentation has been a long-standing problem [1-3].
>
> Specific to SA3D, the geometry information is utilized through the incorporation of a negative refinement (regularization) term in the projection loss (Eq. 5). When SAM overlooks small holes and gaps, the mask passes through these regions and gets projected onto the background behind the object. However, with a viewpoint switched, these inaccurately-segmented regions shift from being behind the target object to the side. In these new views, SAM's foreground prediction no longer includes these regions. Consequently, the mask confidence score for these regions is effectively suppressed by the negative refinement (regularization) term. We hope this explanation can help clear up the confusion.
>
> Presently, we cannot find a dataset that supports evaluation for both fine-grained segmentation and NeRF reconstruction. Consequently, we are unable to provide quantitative metrics to validate the statement. We believe that cultivating such a dataset or benchmark represents a promising avenue for future research.
>
> > Q2: Self prompting strategy: ... the necessity of this particular prompting strategy... How much better than a plain 2D-based NMS? ... yield roughly the same results - if not, how much worse ...?
>
> **A6:** Our self-prompting strategy can be treated as a variant of the NMS algorithm. Without the 3D distance based confidence score decay, it degenerates into a simple 2D NMS, which selects a prompt point with the highest confidence score and then blocks out a surrounding region of it (as shown in Line 159-160). To answer the question, we conducted an ablation experiment on the NVOS dataset:
>
> ||w/ Confidence Decay Term||w/o Confidence Decay Term||
> |-|-|-|-|-|
> |Scene|IoU|Acc|IoU|Acc|
> |fern|82.9|94.4|82.9|94.4|
> |flower|94.6|98.7|94.6|98.7|
> |fortress|98.3|99.7|98.4|99.7|
> |horns_center|96.2|99.3|96.2|99.3|
> |horns_left|90.2|99.4|88.8|99.3|
> |leaves|93.2|99.6|93.2|99.6|
> |orchids|85.5|97.3|85.4|97.3|
> |trex|82.0|97.4|64.0|93.3|
> |mean|90.3|98.2|87.9|97.7|
>
> The above table shows that for most cases, a simple NMS self-prompting is enough. But for hard cases like 'LLFF-trex' (a trex skeleton, as shown in Fig. 5 of our paper), where a large number of depth jumps, the confidence decay contributes a lot. In such a situation, inaccurate masks bleed through gaps in the foreground onto the background. If the self-prompting mechanism generates prompts on these inaccurately-segmented regions, SAM may produce plausible segmentation results that can cheat the IoU-rejection mechanism and finally the segmentation results will involve unwanted background regions.
>
> In addition, performing the decay step only involves some basic matrix calculation, which does not result in big time consumption. Taking the 'LLFF-fern' scene as an example, replacing the self-prompting strategy with a simple NMS only decrease the 27.04 seconds cost to 25.73 seconds.
>
> [1] Couprie, Camille, et al. "Indoor semantic segmentation using depth information." arXiv preprint arXiv:1301.3572 (2013).
>
> [2] Zhang, Zhenyu, et al. "Joint task-recursive learning for semantic segmentation and depth estimation." Proceedings of the European Conference on Computer Vision (ECCV). 2018.
>
> [3] Kerr, Justin, et al. "Lerf: Language embedded radiance fields." arXiv preprint arXiv:2303.09553 (2023).

---

> > ### Comment · Reviewer_Xgwo · 2023-08-21
> >
> > Thanks for the detailed reply - I upgrade my recommendation from weak accept to accept, as I have no issues with the paper based on the rebuttal.

---

> > > ### Author Response · Authors · 2023-08-21
> > > **Thanks for Your Response**
> > >
> > > We sincerely thank you for your response! Your help in reviewing our paper has been very valuable in making it better.

---

### Author Rebuttal · Authors · 2023-08-10

## Common Response for Time Cost Analysis

We thank all reviewers for the insightful comments.

Since several reviewers (Q1 of Reviewer b3JM, W1 & W6 of Reviewer JhYG, W2 of Reviewer Xgwo) express concerns about the time overhead of our method, we discuss this issue here.

We provide per-scene time cost for the LLFF dataset and some unbounded scenes involved in Table 2. Please kindly note that the experiments of Table 1 and Table 2 are both based on the scenes of the LLFF dataset. Consequently, we omit the time cost for Table 1. Besides, the average time cost (seconds) comparisons for each object of the Replica dataset are shown in Table 3. Table 4 supplements time cost analysis regarding the number of views to illustrate the trade-off between time cost and accuracy.

The modified tables and attached table are shown as follows.

---
**Table 1. Comparison with ISRF**

There are three methods involved in the comparison in Table 1: Graph Cut (3D), NVOS and ISRF. The first two methods neither provide the code to reproduce nor clarifies the time overhead in their paper. ISRF provide rough time cost estimations. Thus we compare with ISRF roughly for segmenting an object in a scene:

| ISRF || SA3D | |
|:-:|:-:|:-:|:-:|
| Step |Time Cost | Step |Time Cost |
|||User Intervention (One Time) |-|
|||Initial Segmentation |< 1 second |
| Training feature field | 2.5 minutes |Training Mask Grids | 10 seconds - 3 minutes|
|User Intervention (Many Times)| - || |
| K-Means Clustering | 2 seconds |||
| 3D Feature Query | 1 seconds |||
| Bilateral Region Growing | 0.3 seconds|||

The coarse time cost of ISRF is gathered from their paper. For segmenting an object, ISRF requires many iterations of user intervention (and the following steps). Compared with ISRF, SA3D takes similar or smaller time cost but enjoys a more concise procedure. Please note that for both methods the time cost of pre-training a NeRF is omitted for clear comparisons.

---
**Table 2. Comparisons on LLFF scenes and some 360 degrees scenes**

|Scenes|Single View|||MVSeg|||SA3D (ours)|||
|:-:|:-:|:-:|:-:|:-:|:-:|:-:|:-:|:-:|:-:|
||IoU(%)|Acc(%)|Time Cost(s)|IoU(%)|Acc(%)|Time Cost(s)|IoU(%)|Acc(%)|Time Cost(s)|
|Orchids|79.4|96.0|-|92.7|98.8|264|83.6|96.9|33.9|
|Leaves|78.7|98.6|-|94.9|99.7|271|97.2|99.9|30.1|
|Fern|95.2|99.3|-|94.3|99.2|246|97.1|99.6|26.1|
|Room|73.4|96.5|-|95.6|99.4|284|88.2|98.3|52.7|
|Horns|85.3|97.1|-|92.8|98.7|276|94.5|99.0|75.3|
|Fortress|94.1|99.1|-|97.7|99.7|273|98.3|99.8|46.8|
|Fork|69.4|98.5|-|87.9|99.5|244|89.4|99.6|27.7|
|Pinecone|57.0|92.5|-|93.4|99.2|257|92.9|99.1|73.4|
|Truck|37.9|77.9|-|85.2|95.1|261|90.8|96.7|333.5|
|Lego|76.0|99.1|-|74.9|99.2|263|92.2|99.8|180.9|
|mean|74.6|95.5|-|90.9|98.9|264|92.4|98.9|88.0|

Note that as we introduced in Line 215-217, "Single View" denotes directly projecting the mask of the reference view onto the mask grids without any following operations, which takes almost no time cost. Thus we omit its time cost in Table 2 and Table 3.

---
**Table 3. Comparisons on Replica**

|Methods|metrics|office0|office1|office2|office3|office4|room0|room1|room2|mean|
|:-:|:-:|-|-|-|-|-|-|-|-|-|
|Single View|mIoU(%)|68.7|56.5|68.4|62.2|57.0|55.4|53.8|56.7|59.8|
||Time Cost(s/object)|-|-|-|-|-|-|-|-|-|
|MVSeg|mIoU(%)|31.4|40.4|30.4|30.5|25.4|31.1|40.7|29.2|32.4|
||Time Cost(s/object)|1567|1360|1343|1617|1301|1292|1431|1527|1430|
|SA3D(ours)|mIoU(%)|84.4|77.0|88.9|84.4|82.6|77.6|79.8|89.2|83.0|
||Time Cost(s/object)|81.2|53.9|46.4|62.1|76.4|58.1|61.5|101.2|67.6|

MVSeg does not report the time cost in their paper. We reproduce MVSeg with its official code on the Replica dataset to measure its time cost. As shown in the modified Table 3, MVSeg takes a high time cost as MVSeg needs to train a Semantic-NeRF for each target in its official implementation.

---
**Table 4. Ablation on different numbers of views**

||||||
|-|-|-|-|-|
|Number of Views|5(10%)|9(20%)|21(50%)|43(100%)|
|IoU on Fortress (forward facing)|97.8|98.3|98.3|98.3|
|Time Cost(s)|7.56|12.80|28.98|58.97|
|Number of Views|11(10%)|21(20%)|51(50%)|103(100%)|
|IoU on Lego (360 degrees)|84.5|84.8|91.5|92.2|
|Time Cost(s)|23.49|43.54|103.83|204.93|

Please kindly note that the time cost of SA3D depends on the number of views during the mask training phase. Consequently, there exist differences in the time overhead of SA3D between different scenes. Nonetheless, on the whole, the time overhead of SA3D typically remains within 1 minute.

---
We also provide a possible accelerating solution for future updating. Except for the first reference view, the following optimization can be parallelized. In each iteration, SA3D can select multiple new views and conduct self-prompting and inverse rendering for them simultaneously. Gradients on mask grids can then be gathered for optimization.

We hope the discussion can address the concern of time consumption.

---

### Decision · Program_Chairs · 2023-09-21

**Decision:**

Accept (poster)

**Comment:**

This paper proposes to lift 2D predictions by SAM to the 3D space based on NeRF. The initial reviews are generally positive. The rebuttal further clarified concerns raised by reviewers. In agreement with the reviewers, AC recommends acceptance since the proposed method is simple yet effective.